# IL-1β turnover by the UBE2L3 ubiquitin conjugating enzyme and HECT E3 ligases limits inflammation

Vishwas Mishra [1,3], Anna Crespo-Puig[1,3], Callum McCarthy [1],
Tereza Masonou [1], Izabela Glegola-Madejska[1], Alice Dejoux[1], Gabriella Dow [1],
Matthew J. G. Eldridge [1], Luciano H. Marinelli[1], Meihan Meng[1], Shijie Wang[1],
Daniel J. Bennison [1], Rebecca Morrison[2] & Avinash R. Shenoy [1] ✉

The cytokine interleukin-1β (IL-1β) has pivotal roles in antimicrobial immunity, but also incites inflammatory disease. Bioactive IL-1β is released following proteolytic maturation of the pro-IL-1β precursor by caspase-1. UBE2L3, a ubiquitin conjugating enzyme, promotes pro-IL-1β ubiquitylation and proteasomal disposal. However, actions of UBE2L3 in vivo and its ubiquitin ligase partners in this process are unknown. Here we report that deletion of *Ube2l3* in mice reduces pro-IL-1β turnover in macrophages, leading to excessive mature IL-1β production, neutrophilic inflammation and disease following inflammasome activation. An unbiased RNAi screen identified TRIP12 and AREL1 E3 ligases of the Homologous to E6 C-terminus (HECT) family in adding destabilising K27-, K29- and K33- poly-ubiquitin chains on pro-IL-1β. We show that precursor abundance determines mature IL-1β production, and UBE2L3, TRIP12 and AREL1 limit inflammation by shrinking the cellular pool of pro-IL-1β. Our study uncovers fundamental processes governing IL-1β homeostasis and provides molecular insights that could be exploited to mitigate its adverse actions in disease.

Interleukin-1β (IL-1β) has been at the forefront of cytokine biology for over 50 years, yet the name is a misnomer because IL-1β does much more than 'communicate between leukocytes'[1]. Apart from enhancing the functions of myeloid and lymphoid cells during infection, IL-1β promotes systemic inflammation by activating acute-phase protein production by hepatocytes, prostaglandin release, enhanced coagulation and neutrophil adherence on endothelial cells, and antiviral activity in keratinocytes[1]. In addition, IL-1β is an endogenous pyrogen that induces fever at doses as low as 1 ng.kg⁻¹ body weight in humans. Deregulated IL-1β production drives the pathology of autoinflammatory diseases such as gout, rheumatoid arthritis, diabetes, cardiac and neurodegenerative disease, and hereditary fever syndromes[1]. Given these pivotal roles of IL-1β, we need a better

understanding of processes governing its production and new approaches to prevent its detrimental actions.

IL-1β production is tightly controlled transcriptionally and post-translationally[1,2]. In myeloid cells (e.g., macrophages, monocytes), which are the dominant source of IL-1β, *Il1b* mRNA is induced in an NF-κB−dependent manner following exposure to microbial molecules (e.g., ligands of Toll-like receptors) or cytokines (e.g., TNF)[1]. *Il1b* mRNA codes for the biologically inert pro-IL-1β precursor protein (269 aa; mouse aa numbering). Pro-IL-1β undergoes proteolysis by caspase-1, originally called interleukin-1 converting enzyme, which removes the 'pro' domain (aa 1-117) and triggers the release of the mature, receptor-binding cytokine (aa 118-269). Despite the wealth of information on bioactive IL-1β, regulation of the pro-IL-1β

[1]Medical Research Council Centre for Molecular Bacteriology & Infection, Department of Infectious Disease, Imperial College London, London, UK. [2]Host-Pathogen Interactions in Tuberculosis Laboratory, The Francis Crick Institute, 1 Midland Road, London NW1 1AT, UK. [3]These authors contributed equally: Vishwas Mishra, Anna Crespo-Puig. ✉e-mail: a.shenoy@imperial.ac.uk

precursor prior to its proteolysis by caspase-1 remains poorly understood.

Caspase-1 is activated within inflammasomes, which are multimolecular signalling scaffolds[2]. Inflammasomes detect cytosolic microbial ligands (e.g., cytosolic LPS[3]) or 'sterile' danger signals (e.g., gout-associated uric acid crystals[4]), among other triggers. LPS, a component of Gram-negative bacteria, induces systemic inflammation through its potent ability to induce *Il1b* transcripts and pro-IL-1β conversion via caspase-4/11, NLRP3 (NOD and leucine-rich repeat-containing protein with a pyrin domain 3) and caspase-1. Inflammasome components such as IL-1β, NLRP3, caspase-1, and caspase-11 are thus among the major drivers of septic-shock like disease induced by LPS in mice[5]. Similarly, genetic loss of *Casp1*, *Nlrp3*, or *Casp11* abolishes IL-1β production and neutrophilic inflammatory symptoms induced by sterile signals, such as alum, cholesterol or uric acid crystals[1,2,5].

How cells manage pro-IL-1β abundance or dispose of the precursor cytokine is not completely understood. We previously discovered that the ubiquitin-conjugating enzyme UBE2L3 (also called UBCH7) promotes pro-IL-1β ubiquitylation and proteasomal degradation in macrophages[6]. Ubiquitylation involves the covalent conjugation of ubiquitin to proteins, followed by further additions on one or more lysine residues of ubiquitin to form poly-ubiquitin chains[7,8]. This requires ubiquitin-activating E1 enzymes (2 in mice and humans), ubiquitin-conjugating E2 enzymes (~40 in mice and humans) and ubiquitin E3 ligases (~700 in mice and humans). UBE2L3 is among the most abundant E2 enzymes in cells and only partners with E3 ligases of the Homologous to E6 C-terminus (HECT) and Really Interesting New Gene (RING) between RING (RBR) subfamilies[9,10]. These ubiquitin ligases transfer ubiquitin from the active site cysteine of UBE2L3 to their own catalytic cysteine residue before ubiquitylating substrates[7,8]. UBE2L3-dependent pro-IL-1β ubiquitylation therefore likely involves HECT or RBR ligases, however, the E3 ligases that ubiquitylate pro-IL-1β for its turnover are not known.

In humans, polymorphisms in *UBE2L3* are associated with inflammatory conditions, including arthritis and inflammatory bowel disease, demonstrating a link between UBE2L3 and deregulated inflammation through as yet unknown mechanisms[10]. Silencing UBE2L3 expression enhances pro-IL-1β abundance in macrophages and UBE2L3 overexpression increases pro-IL-1β turnover, indicating a key role for UBE2L3 in IL-1β homeostasis[6]. Furthermore, UBE2L3 is an indirect target of caspase-1 and its own abundance reduces upon inflammasome activation[6]. Whether UBE2L3 abundance is affected by inflammasomes in vivo and the consequence of *Ube2l3*-deletion in whole animals has not been tested.

Here we describe the generation and characterisation of conditional tissue-specific deletion of *Ube2l3* in mice (*Ube2l3ΔMac*), which display elevated pro-IL-1β abundance, heightened secretion of IL-1β and inflammatory disease following inflammasome activation. Through unbiased RNA interference (RNAi) screening, we also identify two HECT-type E3 ligases, TRIP12 and AREL1, that promote pro-IL-1β ubiquitylation and proteasomal turnover. These findings provide insights on the regulation of a potent proinflammatory cytokine and E3 ligases that could be exploited in designing novel therapeutics in the future.

## Results

### Inflammasome activation depletes macrophage UBE2L3 in vivo

Inflammasome activation in human and mouse macrophages triggers the proteasomal degradation of UBE2L3, which results in elevated IL-1β secretion in vitro[6]. To assess UBE2L3 depletion by inflammasomes in vivo, we treated mice with LPS or LPS and ATP, which resulted in IL-1β secretion in the peritoneal lavage (Fig. 1a), confirming inflammasome activation. Flow cytometric analyses of peritoneal macrophages from these mice revealed fewer F4/80+ve macrophages staining positive for UBE2L3 (Fig. 1b, c), and a reduction in the geometric mean

fluorescence intensity for UBE2L3 (Fig. 1d, e). Consistent with this, immunoblots showed lower UBE2L3 abundance in peritoneal macrophages from mice given LPS and ATP (Fig. 1f). These results together indicated that UBE2L3 abundance reduces upon inflammasome activation in vivo. Furthermore, the frequency of UBE2L3+ve peritoneal macrophages inversely correlated with IL-1β levels in the peritoneal lavage fluid (Pearson's correlation coefficient r = −0.7, n = 15, P value: 0.0039; Fig. 1g). Taken together, these results from experiments in vivo and previous findings in vitro[6] are consistent with UBE2L3 acting as a negative regulator of IL-1β production.

### Elevated IL-1β and disease severity in *Ube2l3ΔMac* mice after inflammasome activation

To investigate the functions of UBE2L3 in vivo, we generated a tissue-specific conditional deletion strain because *Ube2l3* is an essential gene in mice[11]. Mice with a 'floxed' *Ube2l3* exon 1 (*Ube2l3fx/fx* mice) were crossed with Csf1r-cre/Esr1 mice that express the Cre recombinase-oestrogen receptor fusion protein whose activity is induced by tamoxifen in Csf1r+ve cells (i.e., macrophages and monocytes, hereafter called *Ube2l3ΔMac* mice; Fig. 2a)[12]. Inducible recombination was verified in primary bone marrow-derived macrophages (BMDMs) treated with 4-hydroxytamoxifen (hTam) as assessed by PCR (Fig. S1A), which led to reduced UBE2L3 expression (Fig. S1B). Similarly, in vivo efficacy of oral tamoxifen administration was verified by the reduced UBE2L3 staining in Csf1r+ve peritoneal macrophages by flow cytometry (Figs. 2b, S1C) and immunoblots (Fig. S1D).

Having established the conditional deletion of *Ube2l3*, we tested three established models of inflammasome−dependent inflammation and quantified IL-1β production and inflammatory parameters (Fig. 2c). In the low-dose LPS model, *Ube2l3ΔMac* mice produced distinctly higher IL-1β in the peritoneal lavage as compared to wildtype animals (Fig. 2d). Secretion of the inflammasome-independent IL-6 cytokine was similar in both groups of mice (Fig. 2e), indicating a lack of effect on an inflammasome-independent cytokine upon deletion of *Ube2l3*.

High-dose of LPS serves as a model of severe Gram-negative bacterial infection-associated sepsis and shock-like inflammatory symptoms driven by inflammasomes. *Ube2l3ΔMac* mice given high-dose of LPS had markedly higher serum IL-1β compared to wildtype mice (Fig. 2f); IL-6 levels remained unchanged (Fig. 2g). In agreement with high IL-1β levels, endotoxic shock-like disease activity scores were prominently higher in *Ube2l3ΔMac* mice (Fig. 2h). These experiments together established a critical role for macrophage UBE2L3 in suppressing IL-1β production and dampening inflammatory disease symptoms in vivo.

We also tested whether UBE2L3 is involved in supressing low-grade sterile inflammation in vivo. Sterile monosodium uric acid crystals (MSU) elicit NLRP3-dependent IL-1β release, which attracts neutrophils into the peritoneum[4]. As compared to wildtype mice injected with MSU, *Ube2l3ΔMac* mice had higher peritoneal IL-1β (Fig. 2i) and neutrophil influx (Figs. 2j, S1E), which further underscore the role of UBE2L3 in being able to restrain IL-1β production in vivo.

Taken together, these experiments establish that conditional deletion of *Ube2l3* in macrophages/monocytes is sufficient to trigger elevated IL-1β, local and systemic inflammation upon inflammasome activation, revealing its important role in limiting IL-1β−driven inflammation.

### *Ube2l3* deletion increases pro-IL-1β protein stability

To examine the effect of *Ube2l3* loss-of-function on pro-IL-1β turnover, we performed cycloheximide (CHX)-chase experiments, in which inhibition of protein translation with CHX reveals the stability of proteins over time ('chase'). Primary peritoneal macrophages from *Ube2l3fx/fx* and *Ube2l3ΔMac* mice were used to assess the turnover of LPS-induced pro-IL-1β, which revealed markedly slower pro-IL-1β protein turnover in *Ube2l3ΔMac* macrophages (Fig. 3a). Pro-IL-1β turnover

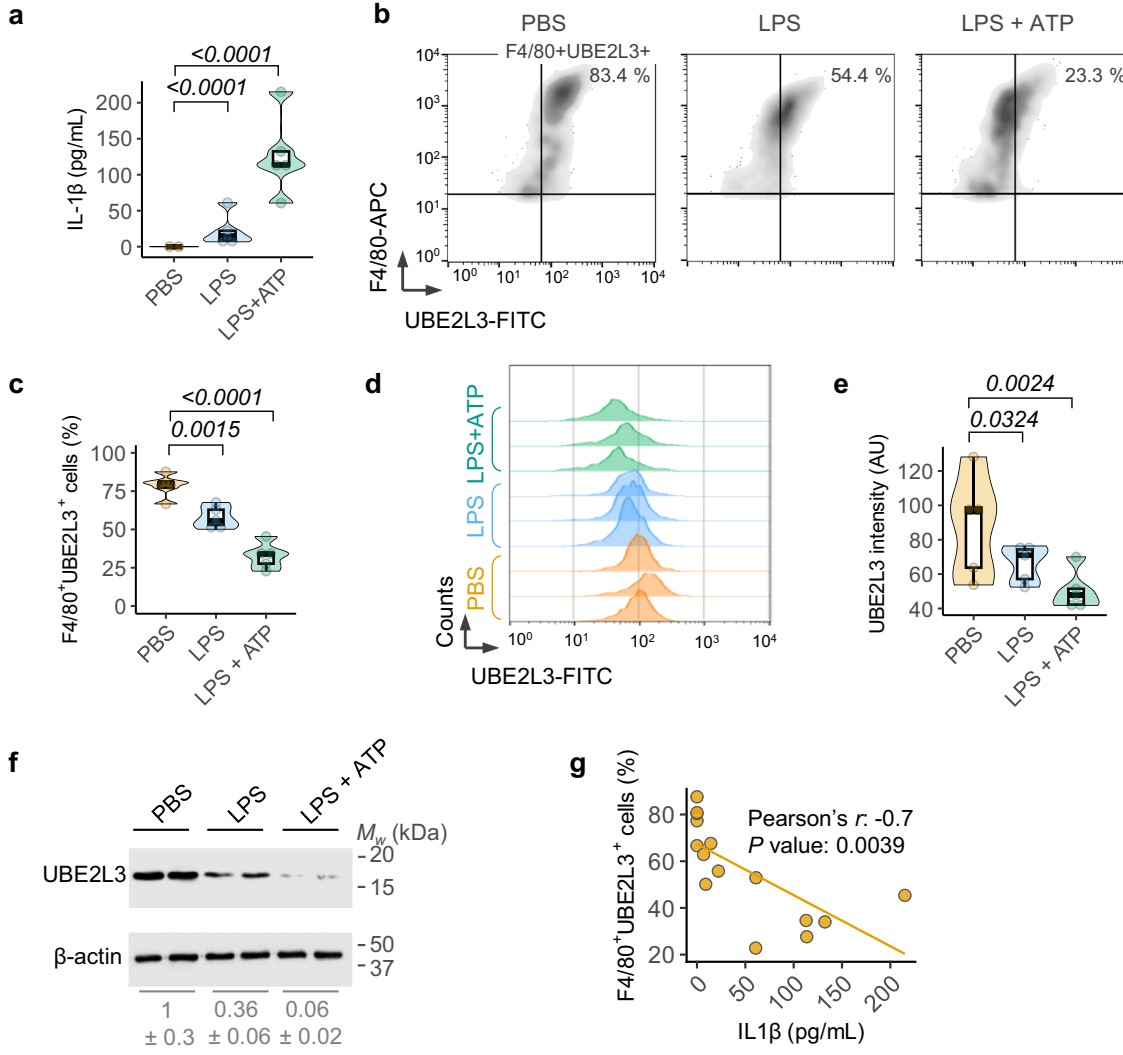

**Fig. 1 | UBE2L3 is depleted in vivo upon inflammasome activation.**
**a** Quantification of IL-1β by ELISA of peritoneal lavage from mice given PBS, LPS (5 μg, 3 h) or LPS + ATP (5 μg, 3 h + 50 μmol, 10 min) intraperitoneally. **b, c** Representative flow cytometry density plots (**b**) of peritoneal cells stained with F4/80 (macrophage surface marker) and intracellular UBE2L3 and quantification (**c**) of the percentage double-positive cells from mice given PBS, LPS or LPS + ATP as described in **a**. **d, e** Representative histograms of counts of F4/80 and UBE2L3 double-positive cells (**d**) and the graph of mean geometric fluorescence intensity of UBE2L3 in F4/80+ve cells (**e**). Mice were treated as in **a**. AU = arbitrary units. **f** Representative immunoblots of UBE2L3 and β-actin in lysates from peritoneal macrophages. Each lane represents an individual mouse given PBS, LPS or LPS +

ATP (5 μg for 3 h + 50 μmol 10 min) intraperitoneally as labelled. Numbers below (mean ± SD) are UBE2L3/β-actin ratios relative to the PBS samples. Data from one of three similar experiments. **g** Plot showing negative correlation between IL-1β measured in peritoneal lavage fluid and percentage of UBE2L3+ve peritoneal macrophages from the same mouse by flow cytometry. Each dot represents a mouse given PBS, LPS or LPS + ATP as in **a–e**. Each lane in **d, f**, and dot in **a, c, e, g** represents an individual mouse; $n = 5$ mice for experiments in **a–e**. Data distribution is depicted with violin, box (25th to 75th percentile, line at median), and whiskers (± 1.5 x IQR). Two-tailed $P$ value for indicated comparison from correlation analysis (**g**) or mixed effects ANOVA (**a, c, e**).

required proteasomal activity, as the proteasome inhibitor MG132 blocked pro-IL-1β clearance (Fig. 3a). Similar experiments revealed slower pro-IL-1β turnover in *Ube2l3^ΔMac* BMDMs (Fig. S1F). We estimated that the half-life of pro-IL-1β increased significantly from 3 h in wild-type macrophages to 5 h in *Ube2l3*-deleted macrophages (Fig. 3b), indicating that pro-IL-1β is more stable in the absence of *Ube2l3*. Importantly, LPS-induced *Il1b* mRNA measured by qRT-PCR was similar in both genotypes (Fig. 3c), which verified that UBE2L3 controls pro-IL-1β abundance post-translationally. Taken together, we conclude that UBE2L3 is required for the proteasomal turnover of pro-IL-1β.

In line with high pro-IL-1β levels, NLRP3 inflammasome activation with LPS and nigericin resulted in prominently higher mature IL-1β as measured by immunoblots and ELISA in *Ube2l3^ΔMac* primary BMDMs (Fig. 3e, f). Importantly, caspase-1 activation into its p20 form and gasdermin-D cleavage were similar in both genotypes (Fig. 3e), ruling

out impacts on inflammasome activation or proteolysis of another key substrate of caspase-1. In agreement with this, IL-6 release and pyroptotic cell death as measured by propidium iodide dye uptake were also indistinguishable between the two genotypes (Fig. 3g, h). These results are indicative of normal NF-κB signalling, inflammasome priming and activation and pyroptotic cell death in *Ube2l3^ΔMac* cells. We therefore conclude that deletion of *Ube2l3* reduces pro-IL-1β protein turnover but, it does not affect inflammasome activation or pyroptosis. Taken together, these results from knockout mice unequivocally establish a crucial role for UBE2L3 in acting independently of inflammasome activation in pro-IL-1β turnover.

**RNAi screen for E3 ligases that promote pro-IL-1β turnover**
What is the mechanism underlying UBE2L3-dependent pro-IL-1β turnover? As UBE2L3 is a ubiquitin-conjugating enzyme, we sought to

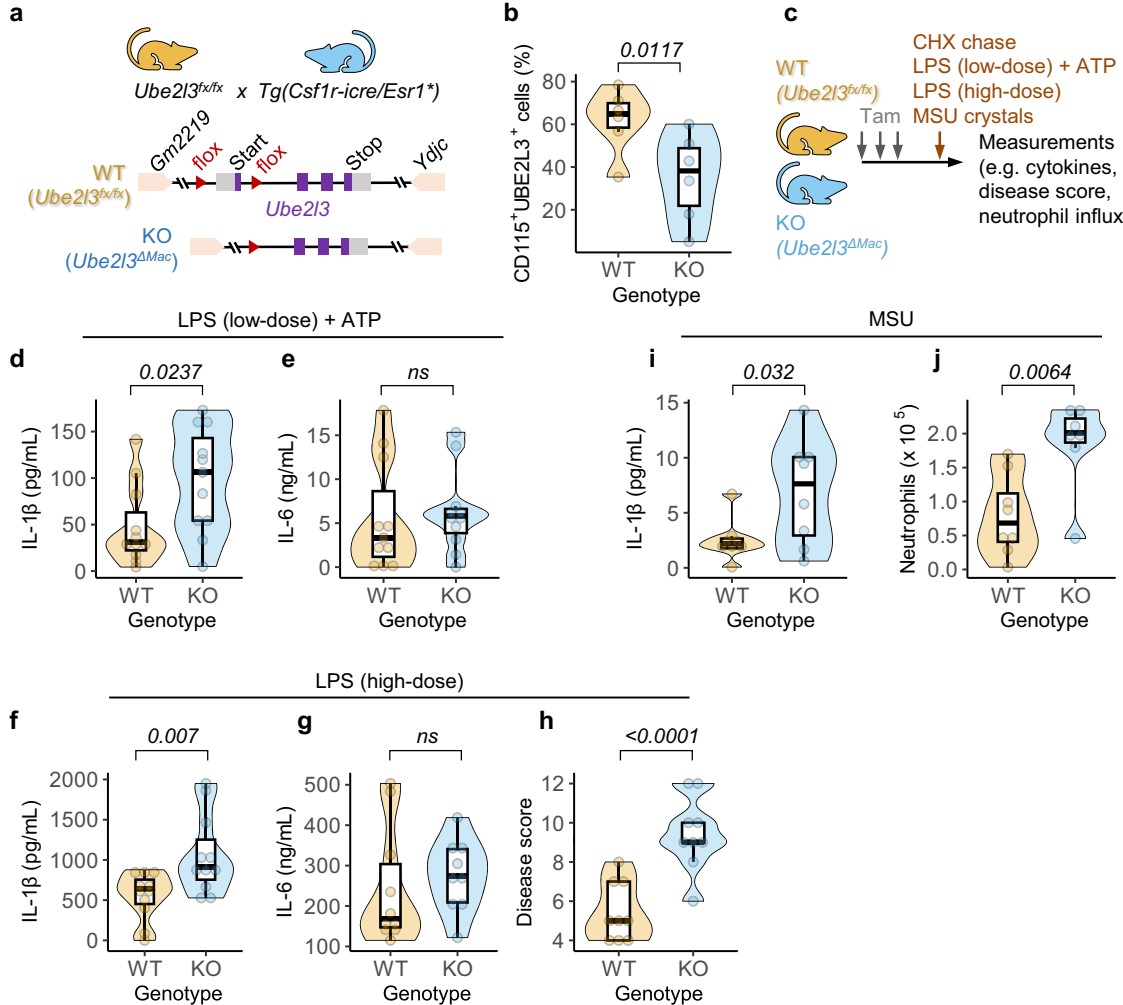

**Fig. 2 | Elevated IL-1β in *Ube2l3^ΔMac* mice after inflammasome activation.**
**a** Schematic depiction of the strategy to generate conditional, macrophage-specific deletion of *Ube2l3*. Exon 1 of *Ube2l3* was 'floxed', leading to its deletion by Cre recombinase whose expression is controlled by the Csf1r (also called CD115) promoter and activity is induced by oral administration of tamoxifen in vivo or 4-hydroxytamoxifen in vitro in macrophages. **b** Flow cytometric analysis showing percentage of Csf1r/CD115 and UBE2L3 +ve peritoneal macrophages isolated from mice of the indicated genotypes given tamoxifen orally on 3 consecutive days. WT = *Ube2l3^fx/fx*; KO = *Ube2l3^ΔMac*. **c** Schematic depiction of inflammatory models tested in *Ube2l3^fx/fx* (WT) and *Ube2l3^ΔMac* (KO) mice. Mice were given tamoxifen (Tam; 80 mg.kg⁻¹) via oral gavage on 3 consecutive days, and the indicated inflammatory stimuli intraperitoneally on day 5. Low dose LPS + ATP = 5 µg LPS per mouse for 3 h followed by ATP (50 µmol) for 10 min. High dose LPS = 30 mg.kg⁻¹ LPS for 3 h. MSU

crystals = monosodium uric acid crystals (1 mg per mouse) for 6 h. See Methods for details. **d, e** Quantification of IL-1β (**d**) and IL-6 (**e**) by ELISA of peritoneal lavage from mice of the indicated genotypes following low dose LPS model. WT = *Ube2l3^fx/fx*; KO = *Ube2l3^ΔMac*. **f–h** Quantification of IL-1β (**f**) and IL-6 (**g**) by ELISA of serum, and disease scores (**h**), of mice of the indicated genotypes following high-dose LPS treatment. WT = *Ube2l3^fx/fx*; KO = *Ube2l3^ΔMac*. **i, j** Quantification of IL-1β (**i**) by ELISA and flow cytometry-based counts of neutrophils (**j**) in the peritoneal lavage of mice injected with MSU crystals. WT = *Ube2l3^fx/fx*; KO = *Ube2l3^ΔMac*. Each dot in **b, d–j** represents a mouse. Number of mice as follows: **b**, $n = 6$; **d-e**, $n = 11$; **f–h**, $n = 9$; **i, j**, $n = 8$. Data distribution is depicted with violin, box (25th to 75th percentile, line at median), and whiskers (±1.5 x IQR); data are pooled from 2-4 independently repeated experiments. Two-tailed $P$ values for indicated comparisons from mixed-effects ANOVA. ns – not significant ($P > 0.05$).

identify the ubiquitin E3 ligase(s) that it co-opts for pro-IL-1β ubiquitylation. To narrow down the potential E3 ligases that could be involved in this process, we measured pro-IL-1β abundance in the absence and presence of broad-specificity inhibitors of subclasses of E3 ligases alongside MG132 as a positive control. Firstly, BAY-11-7082, which can inhibit UBE2L3 and RBR E3 ligases within the linear ubiquitin assembly complex (LUBAC)[13], among other targets including IKK, increased pro-IL-1β abundance as compared to vehicle (DMSO)-treated cells (Fig. S2A). Treatment with MLN4924 (which blocks Cullin-RING superfamily of E3 ligases[14]) did not affect pro-IL-1β abundance (Fig. S2A), ruling out the involvement of Cullin-RING E3 ligases. Heclin, an inhibitor of the HECT-type E3 ligases NEDD4, SMURF2 and WWP1[15], also did not affect pro-IL-1β abundance (Fig. S2A). Based on these results we designed a family-wide siRNA screen against 43 HECT and RBR E3 ligases in pro-IL-1β clearance.

Pro-IL-1β expression is induced by NF-κB activation by TLR ligands (e.g., LPS) or proinflammatory cytokines (e.g., TNF) through signalling steps that involve ubiquitylation/deubiquitylation (e.g., LUBAC, TRAFs, A20)[1,2]. As we wanted to specifically measure changes in pro-IL-1β protein abundance over time, we engineered a doxycycline-inducible pro-IL-1β expression system that is independent of NF-κB signalling and transcription of the native *Il1b* transcript. Doxycycline concentration-dependent expression of pro-IL-1β^Strep (pro-IL-1β with two C-terminal StrepTag II tags; Fig. S2B, C) and its abrogation by silencing *rtta3*, the transcription factor that governs doxycycline-inducible expression (Fig. S2B) confirmed the specificity of the engineered expression system. We also verified that the mechanism and kinetics of pro-IL-1β^Strep turnover were similar to that of endogenous pro-IL-1β in several independent ways. Firstly, the abundance of LPS-induced endogenous pro-IL-1β and doxycycline-induced pro-IL-1β^Strep

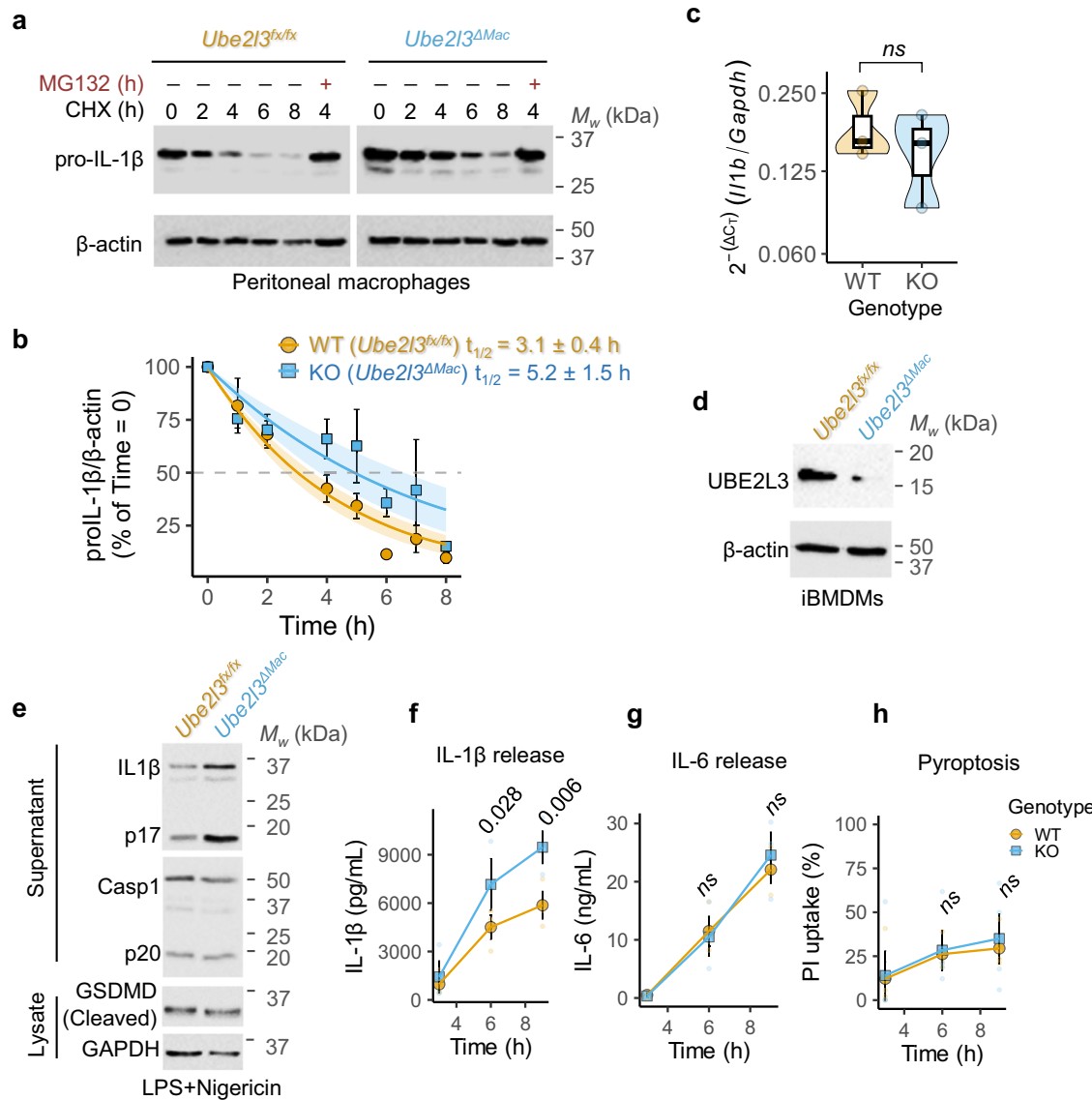

**Fig. 3 | *Ube2l3* deletion increases pro-IL-1β stability in macrophages.**
**a** Representative immunoblots of pro-IL-1β and β-actin in cell lysates from cyclo-heximide (CHX)-chase experiments on peritoneal macrophages from mice of indicated genotypes given tamoxifen to induce *Ube2l3*-deletion in vivo. Cells were treated with LPS (250 ng.mL$^{-1}$) for 14 h followed by CHX (10 μg.mL$^{-1}$) for the indicated times. MG132 (10 μM) was added for the last 3 h in the indicated samples.
**b** Rates of turnover of pro-IL-1β in WT (*Ube2l3$^{fx/fx}$*) and KO (*Ube2l3$^{ΔMac}$*) macro-phages. Immunoblots of CHX-chase experiments as described in **a** in peritoneal macrophages and primary BMDMs (Fig. S1F) were quantified, and the percentage of pro-IL-1β normalised to β-actin relative to Time = 0 h is plotted. Half-live (t$_{1/2}$) ± 95 % confidence interval (CI) are indicated; shaded regions indicate 95 % CI of the fit.
**c** Relative expression of *Il1b* mRNA normalised to *Gapdh* in peritoneal macrophages isolated from WT (*Ube2l3$^{fx/fx}$*) and KO (*Ube2l3$^{ΔMac}$*) mice. Cells were treated with LPS (250 ng.mL$^{-1}$) for 14 h. **d** Representative immunoblots of UBE2L3 and β-actin in cell

lysates of iBMDMs of the indicated genotypes treated with 4-hydroxytamoxifen (2 μM, 48 h). **e** Representative immunoblots of cell lysates and supernatants of primary BMDMs of the indicated genotypes given 4-hydroxytamoxifen (2 μM, 48 h) followed by LPS (250 ng.mL$^{-1}$, 6 h) and nigericin (50 μM, 1 h) to activate inflam-masomes. **f**–**h** Quantification of secreted IL-1β (**f**) and IL-6 (**g**) by ELISA and pyr-optosis (**h**) in primary BMDMs of the indicated genotypes treated with LPS (250 ng.mL$^{-1}$) for the indicated times followed by nigericin (50 μM, 1 h). Pyroptosis was measured using propidium iodide (PI) dye uptake assays. WT = *Ube2l3$^{fx/fx}$*; KO = *Ube2l3$^{ΔMac}$*. Number of independent experiments (*n*) as follows: **a**, *n* = 2; **b**, *n* = 5; **c**, *n* = 3 mice; **d**, *n* = 4; **e**, *n* = 3; **f**–**h**, *n* = 3. Each dot in **c**, and small dots in **f**–**h** represents an independent repeat. Data distribution is depicted in **c** with violin, box (25th to 75th percentile, line at median), and whiskers ( ± 1.5 x IQR); in **b**, **f**–**h** with mean (large symbol) and SEM error bars. Two-tailed *P* values for indicated com-parisons from mixed effects ANOVA; ns = not significant (*P* > 0.05).

declined similarly over time (Fig. S2D). The same inhibitors that blocked endogenous pro-IL-1β turnover as above also blocked pro-IL-1β$^{Strep}$ turnover, i.e., only inhibitors of the proteasome (MG132) or UBE2L3 (BAY-11-7082) reduced pro-IL-1β$^{Strep}$ levels (Fig. 4a). Most importantly, RNAi-mediated silencing of UBE2L3 expression increased pro-IL-1β$^{Strep}$ abundance (Fig. 4b). These results together endorsed the doxycycline-inducible pro-IL-1β$^{Strep}$ as a faithful reporter for our RNAi screen.

UBE2L3 is a unique E2 enzyme that can only transfer the ubiquitin from its active-site to cysteine residues in the active sites of E3 ligases,

and unlike other E2 enzymes it cannot discharge ubiquitin from its active site directly to lysine residues on target proteins[9]. Therefore, it only cooperates with HECT and RBRs E3 ligases which ubiquitylate proteins by first transferring ubiquitin to their own active site cysteine residue; in contrast, RING ligases directly transfer ubiquitin from their cognate E2 to lysine on substrates. Therefore, family-wide siRNA screening was performed against 43 E3-ligases of the HECT and RBR families, which to identify E3 ligases involved in pro-IL-1β turnover (Fig. 4c; see Methods). Wells treated with MG-132 served as positive controls ('high' pro-IL-1β$^{Strep}$ levels), and wells given siRNA against *rtta3*

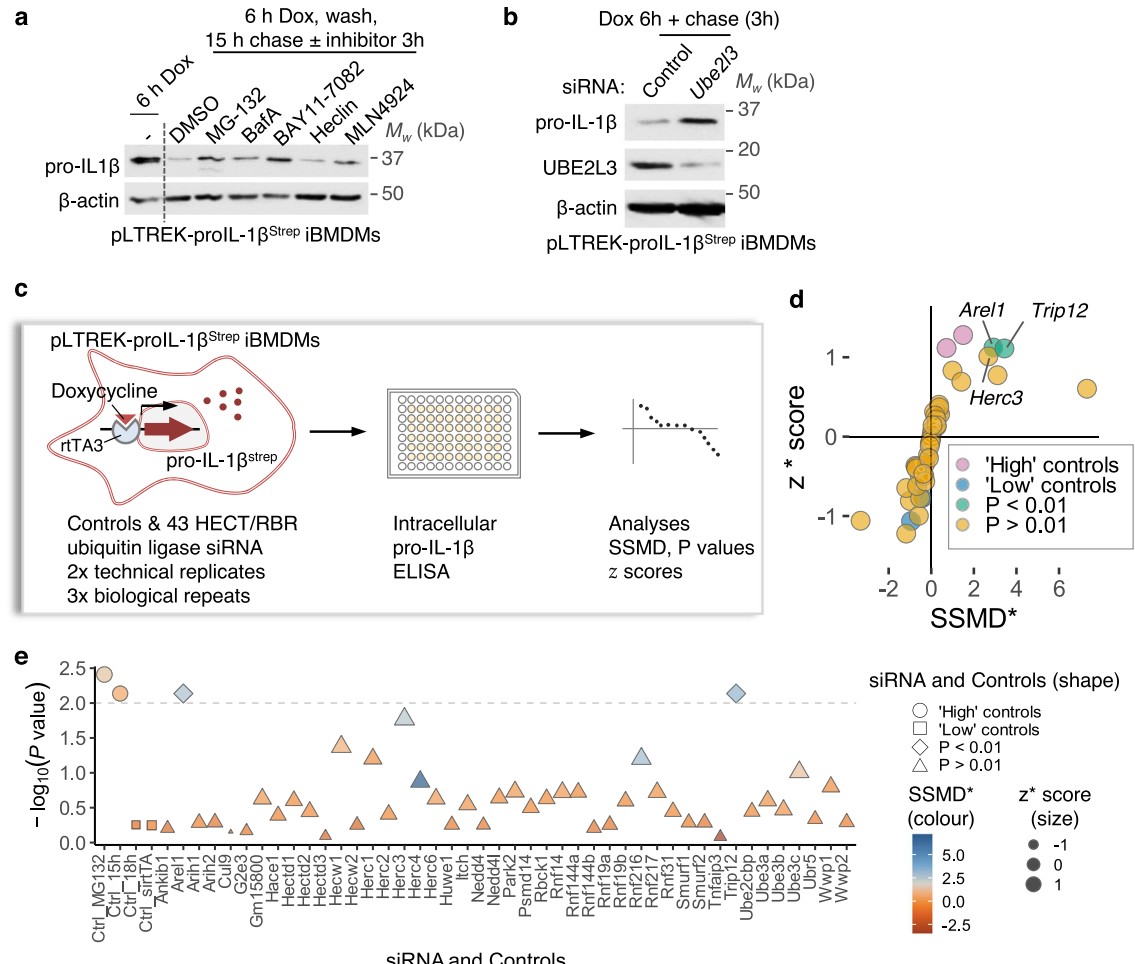

**Fig. 4 | siRNA screen identifies the E3 ligases TRIP12 and AREL1 in promoting pro-IL-1β turnover. a** Representative immunoblots of cell lysates from iBMDMs stably expressing doxycycline (Dox)-inducible proIL-1β$^{Strep}$ treated with the indicated inhibitors. Cells were treated with Dox (500 ng.mL$^{-1}$) for 6 h, washed, and incubated for 18 h ('chased'); inhibitors or the vehicle DMSO was added for the last 3 h. Cell lysate collected at 6 h (sample labelled (−)) served as a control.
**b** Representative immunoblots showing the effect of silencing *UBE2L3* on the abundance of Dox-induced pro-IL-1β$^{Strep}$. pLTREK-proIL-1β$^{Strep}$ iBMDMs were transfected with non-targeting Control or *UBE2L3* siRNA for 72 h, followed by Dox treatment (500 ng.mL$^{-1}$) and a 3 h chase as indicated. **c** Schematic depiction of the siRNA screen against 43 HECT and RBR E3 ligases in pLTREK2-pro-IL-1β$^{Strep}$ iBMDMs. Cells in 96-well plates were screened independently repeated $n = 3$ times with technical duplicates within each repeat. Cells were transfected with siRNA for 72 h,

treated with Dox, and intracellular pro-IL-1β$^{Strep}$ was quantified by ELISA. z* score, SSMD* and *P* values were calculated as described in Methods. **d** Summary plot from siRNA screen described in **c**. Mean z* score and SSMD* from three independent repeats are plotted, with controls and siRNA depicted in different colours as indicated in the legend. **e** Results from the siRNA in screen described in **c**. Size of the symbols indicates mean of z* scores from three independent repeats. FDR-adjusted *P* values for comparison for each of the siRNA with the 'Ctrl_18 h' group are plotted after a -log$_{10}$ transformation. Shapes of the symbols denote controls and *P* value cut-off (indicated with a dotted line at $P = 0.01$). The colour scheme indicates divergent SSMD* scores. *Trip12* and *Arel1* scored above thresholds. Images in **a**, **b** represent experiments done $n = 3$ times. Each dot in **d**, **e** represents one gene or control conditions in the siRNA screen, which was independently repeated $n = 3$ times.

as negative controls ('low' pro-IL-1β$^{Strep}$ levels; Fig. 4d, e). Three independent RNAi screens quantified intracellular pro-IL-1β leading to the identification of two genes that markedly increased pro-IL-1β$^{Strep}$ abundance upon their silencing. These were two HECT-type E3 ligases: TRIP12 and AREL1 (Fig. 4d, e).

### TRIP12 and AREL1 promote pro-IL-1β turnover and reduce mature IL-1β production
We next directly tested whether the E3 ligases identified in the screen regulated endogenous pro-IL-1β. In agreement with results from the screen, the abundance of LPS-induced pro-IL-1β increased upon silencing TRIP12 or AREL1 as measured by immunoblots and ELISA of intracellular pro-IL-1β (Figs. 5a, b, S2E−G); *Il1b* transcription remained unaffected by *Trip12* or *Arel1* silencing (Fig. 5c), showing that these proteins do not broadly affect NF-κB−dependent cytokine transcription. Silencing *Trip12* and *Arel1* together increased pro-IL-1β

abundance to a greater extent than silencing each ligase alone, indicating that each protein partially contributes to destabilising pro-IL-1β (Fig. S2H). Notably, silencing the next potential 'hit' based on SSMD* score alone, *Herc3*, had no effect on pro-IL-1β abundance (Fig. 5a, b, S2E), underscoring that the high stringency cut-off in the screen was appropriate.

We next asked whether TRIP12 and AREL1 interacted with each other and UBE2L3. Due to the lack of suitable antibodies against mouse TRIP12 and AREL1 for coimmunoprecipitation of endogenous proteins and the failure of antibodies against the human proteins in these experiments, we took advantage of transient transfections in HEK293E cells to assess their interactions. Indeed, endogenous UBE2L3 as well as transfected UBE2L3$^{His}$ (with a C-terminal hexa-histidine tag) interacted with TRIP12 and AREL1 (Fig. 5d). Furthermore, TRIP12 and AREL1 also formed a stable complex (Fig. 5e), suggesting that they could act together in the same pathway for pro-IL-1β turnover. To test this we

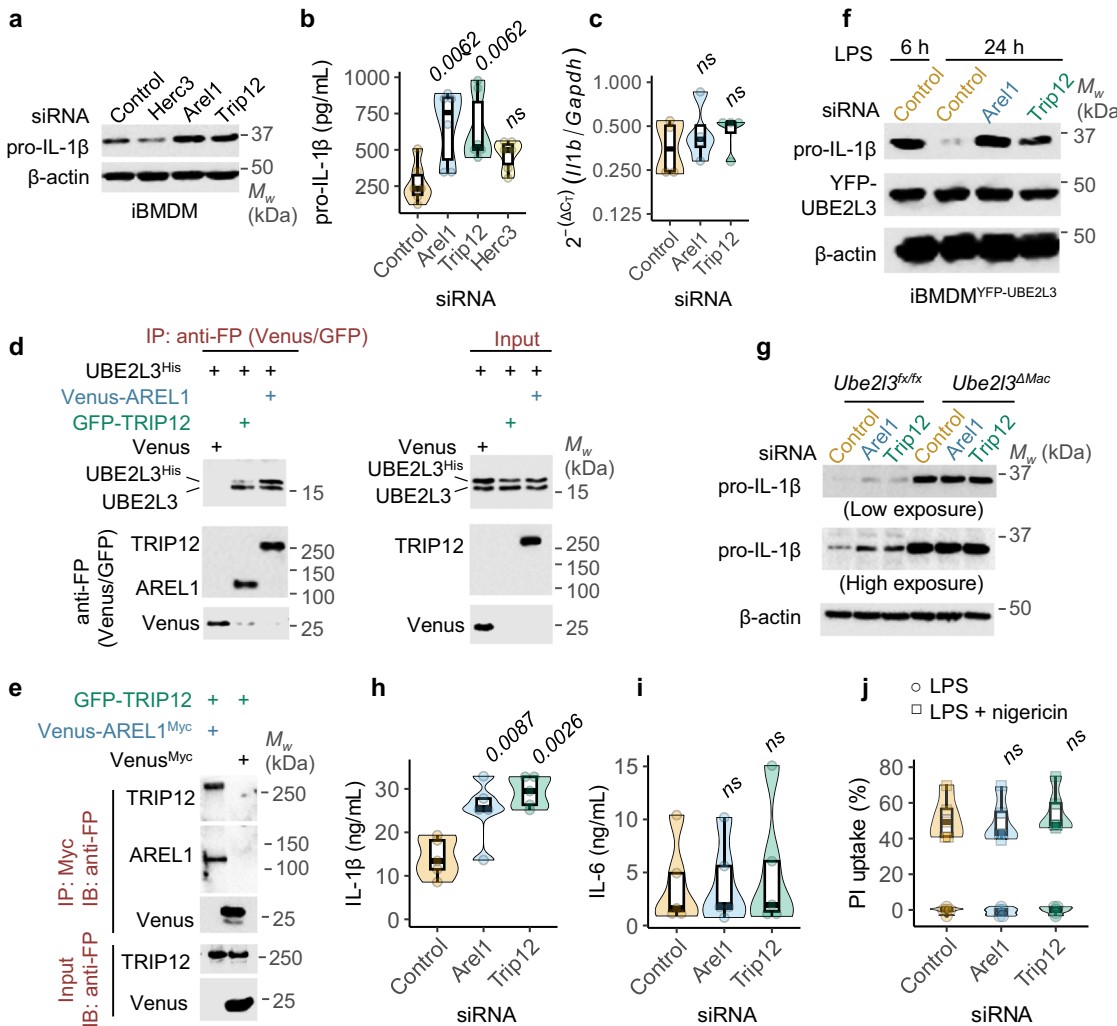

**Fig. 5 | TRIP12 and AREL1 interact with UBE2L3, promote pro-IL-1β clearance and suppress mature IL-1β production. a–c** Increased abundance of endogenous pro-IL-1β upon *Trip12* or *Arel1* silencing. iBMDMs were transfected with non-targeting Control or the indicated siRNA for 72 h followed by treatment with LPS (250 ng.mL⁻¹) for 18 h. Representative immunoblots for endogenous pro-IL-1β (**a**), quantification of intracellular pro-IL-1β by ELISA (**b**) and relative expression of *Il1b* transcript normalised to *Gapdh* (**c**) are shown. **c** Representative images from immunoprecipitation (IP) and immunoblot (IB) experiments showing the inter-action of endogenous UBE2L3 and transiently expressed UBE2L3^His with AREL1 and TRIP12 in HEK293E cells. Anti-FP (fluorescent protein) antibody detects both GFP and Venus tags. **e** Representative images from IP and IB experiments showing the interaction between AREL1 and TRIP12 upon transient expression of the indicated proteins in HEK293E cells. Anti-FP (fluorescent protein) antibody detects both GFP and Venus tags. **f** Representative immunoblots from experi-ments assessing the abundance of pro-IL-1β in lysates of iBMDMs stably

expressing YFP-UBE2L3. Cells were transfected with nontargeting Control or the indicated siRNA for 72 h and then treated with LPS (250 ng.mL⁻¹) for the indicated times. **g** Representative immunoblots from experiments assessing the abundance of pro-IL-1β in lysates from of iBMDMs given non-targeting Control or the indi-cated siRNA for 72 h followed by treatment with LPS (250 ng.mL⁻¹) for 24 h. **h-j** Quantification of IL-1β (**h**) and IL-6 (**i**) in supernatants by ELISA and pyroptosis (**j**) of iBMDMs given the indicated siRNA for 72 h, followed by inflammasome acti-vation with LPS (250 ng.mL⁻¹, 3 h) and nigericin (50 μM, 1 h). Images represent independent experiments (*n*) as follows: **a, f, g**, *n* = 3; **d, e**, *n* = 2. In **b-c, h–j** each dot represents a biologically independent experiment (*n* = 6 in **b**, *n* = 4 in **c**, *n* = 5 in **h–j**). Data distribution is depicted with violin, box (25th to 75th percentile, line at median), and whiskers (± 1.5 x IQR). Two-tailed P values for comparisons with samples given Control siRNA in **b-c, h-j** from mixed-effects ANOVA. ns – not significant (*P* > 0.05).

used UBE2L3 gain-of-function (iBMDMs overexpressing UBE2L3) or loss-of-function (*Ube2l3^ΔMac* cells). Overexpression of UBE2L3 in mac-rophages (iBMDM^YFP-UBE2L3) increases pro-IL-1β ubiquitylation and turnover[6]. Silencing *Trip12* or *Arel1* blocked the accelerated loss of pro-IL-1β protein seen with UBE2L3 overexpression (Fig. 5f), which indi-cates that these E3 ligases are involved in UBE2L3-driven rapid clear-ance of pro-IL-1β. In a complementary experiment, we silenced *Trip12* or *Arel1* in *Ube2l3^ΔMac*, which resulted in no further increase in pro-IL-1β abundance (Fig. 5g). This indicates that UBE2L3 is critical for pro-IL-1β turnover and TRIP12 and AREL1 E3 ligases cannot enlist alternative ubiquitin E2 conjugating enzymes for this process. Taken together, we conclude that TRIP12 and AREL1 form a complex containing UBE2L3 to promote pro-IL-1β turnover.

We reasoned that like with loss of *Ube2l3*, elevated pro-IL-1β abundance upon silencing TRIP12 or AREL1 would lead to increased IL-1β release by inflammasomes. Indeed, macrophages given siRNA against *Trip12* or *Arel1* released markedly higher IL-1β following NLRP3 inflammasome activation with LPS and nigericin as compared to cells given non-targeting Control siRNA (Fig. 5h); as expected, the release of IL-6 remained unchanged (Fig. 5i). Pyroptotic membrane damage measured by propidium iodide dye uptake assay was also unaffected (Fig. 5j), indicating similar levels of caspase-1 inflammasome activity that drives cell death. These results led us to conclude that like UBE2L3, TRIP12 and AREL1 do not affect inflammasome priming or activation; they limit IL-1β production by reducing the cellular pool of pro-IL-1β available for caspase-1−mediated proteolytic maturation.

## Pro-IL-1β is ubiquitylated by TRIP12 and AREL1

We next investigated whether TRIP12 or AREL1 interacted with pro-IL-1β. Experiments revealed that pro-IL-1β specifically co-immunoprecipitated with TRIP12 and AREL1 (Fig. S3A), which indicates that these proteins can form stable complexes. To address whether TRIP12 or AREL1 could stimulate pro-IL-1β ubiquitylation we generated HEK293E cells stably expressing pro-IL-1β[His] with C-terminal hexa-histidine tag that allowed pull-downs under denaturing conditions (to remove non-covalent protein-protein interactions while retaining covalently attached ubiquitin chains) with immobilised metal affinity chromatography. We co-transfected plasmids encoding HA-tagged wildtype ubiquitin along with either TRIP12 or AREL1, or Venus as negative control, followed by denaturing pull-down and anti-HA immunoblots. This revealed marked ubiquitylation of pro-IL-1β by both TRIP12 and AREL1 as seen by 'ubiquitin smears' in western blots (Figs. 6a, b, S3B, C).

We next asked which poly-ubiquitin chain types were added on to pro-IL-1β by TRIP12 and AREL1. To address this question, we used similar experiments as above, but with HA-tagged ubiquitin variants with a single K → R mutation at each lysine residue (K6R, K11R, K27R, K29R, K33R, K48R and K63R) to abrogate poly-ubiquitin chains of those types. Denaturing Ni-NTA pull-downs of pro-IL-1β[His] immunoblotted with anti-HA antibodies revealed reduced ubiquitylation by TRIP12 with Ubi-K27R, Ubi-K29R or Ubi-K33R variants (Figs. 6a, S3B), which suggested that TRIP12 adds K27-, K29- and K33-ubiquitin chains on pro-IL-1β. Similar experiments with AREL1 revealed reduced pro-IL-1β ubiquitylation in cells containing Ubi-K27R and Ubi-K33R, indicating that AREL1 preferentially added K27- and K33-ubiquitin chains (Figs. 6b, S3C). Taken together, we conclude that TRIP12 and AREL1 E3 ligases target pro-IL-1β for ubiquitylation with K27-, K29- and K33- poly-ubiquitin chains.

## TRIP12 and AREL1 ubiquitylate pro-IL-1β in the 'pro' domain

Which residues in pro-IL-1β are preferentially ubiquitylated by TRIP12 and AREL1? Previously, Lys 133 in the mature cytokine region was reported to be ubiquitylated[16–18], however, a K133R mutant is still ubiquitylated suggesting that other sites, including in the 'pro' domain, may exist[16–18]. To test whether 'pro' domain lysines were sites for ubiquitylation, we mutated all four lysines to arginine to generate pro-IL-1β-4R (K30R/K32R/K58R/K72R; Fig. 6c). We co-transfected HEK293E cells with plasmids encoding pro-IL-1β variants and either TRIP12 or AREL1 or Venus as negative control, and enriched all ubiquitylated proteins by pull-downs with recombinant hexahistidine-tagged trypsin-resistant tandem ubiquitin interacting entities ([His]TUBE, Fig. S3D) followed by immunoblotting against pro-IL-1β (referred to as TUBE assays)[19,20]. TUBE assays revealed a drastic reduction in ubiquitylation of pro-IL-1β-4R by TRIP12 and AREL1 as compared to wildtype pro-IL-1β (Fig. 6d, e). These results demonstrated that the 'pro' domain is a major site for ubiquitylation. Consistent with ubiquitylation at K133, the additional K133R mutation in pro-IL-1β-4R (pro-IL-1β-5R) led to a further reduction in ubiquitylation (Fig. 6d, e). We therefore conclude that the 'pro' domain of pro-IL-1β in addition to K133 contains major sites of ubiquitylation by TRIP12 and AREL1 (Fig. S4).

## Pro-IL-1β ubiquitylation affects its stability but not proteolysis

To assess whether the reduced ubiquitylation of pro-IL-1β mutants affected their turnover, we performed cycloheximide-chase experiments. As expected with the destabilising role of 'pro' domain ubiquitylation, both pro-IL-1β-4R and pro-IL-1β-5R variants were remarkably more stable than wildtype pro-IL-1β, with half-lives of 7.3 h and 16 h, respectively, as compared to 4.3 h for wildtype pro-IL-1β (Figs. 6f, S3E). We next asked whether these variants could be converted to similar extent as wildtype pro-IL-1β by caspase-1. To assess their proteolytic maturation, we co-transfected them in HEK293E cells with plasmids encoding caspase-1 and ASC-CFP, which generates active caspase-1 due to spontaneous oligomerisation of CFP.

Proteolysis of wildtype pro-IL-1β was similar to that of pro-IL-1β-4R and pro-IL-1β-5R (Fig. 6g), which indicates that these lysine residues do not affect cleavage by caspase-1. Taken together (Fig. S4), we conclude that pro-IL-1β ubiquitylation reduces its stability and when this process is blocked, for example by loss of UBE2L3, TRIP12 or AREL1 or mutation of target lysine residues, more pro-IL-1β accumulates in cells and this pool is susceptible to caspase-1−dependent proteolysis.

## Discussion

Here we showed that UBE2L3 levels inversely correlate with IL-1β production in vivo and demonstrated its role in suppressing IL-1β−driven inflammation by generating and characterising $Ube2l3^{ΔMac}$ mice (Fig. S4). The loss of UBE2L3 in macrophages/monocytes is sufficient for elevated IL-1β and inflammation following inflammasome activation (Fig. S4). Mechanistically, UBE2L3 co-opted the HECT-type E3 ligases TRIP12 and AREL1, which ubiquitylated pro-IL-1β on lysine residues mainly in the 'pro' domain with K27-, K29- and K33- poly-ubiquitin chains. UBE2L3, TRIP12 and AREL1 act independently of inflammasomes to impede inflammation by reducing the cellular pool of the pro-IL-1β precursor available for caspase-1 (Fig. S4).

Myeloid cells are the major source of IL-1β and drivers of disease. For instance, macrophage-specific expression of a constitutively active mutant of NLRP3 ($Nlrp3^{A350V}$) results in overactive inflammasomes, high IL-1β and systemic inflammation[21]. Similarly, mice with macrophage-specific deletion of negative regulators of innate immune signalling, such as NF-κB subunit p65 ($Rela$)[22], IKKB ($Ikbkb$)[23] or TAK1 ($Map3k7$)[24], also display similar elevation in circulating IL-1β (~2-3−fold) as $Ube2l3^{ΔMac}$ mice upon LPS injection. In contrast to these regulators, phagocyte UBE2L3 acts post-transcriptionally as a critical suppressor of IL-1β−driven inflammation.

In humans, $UBE2L3$ polymorphisms are linked to systemic lupus erythematosus, rheumatoid arthritis, juvenile idiopathic arthritis, inflammatory bowel disease, and ulcerative colitis[10]. As UBE2L3 is ubiquitously expressed, its cell type- and context-specific actions need to be further examined to pinpoint disease mechanisms[10]. In addition, bacterial virulence factors, such as NleL from enterohaemorrhagic *E. coli*, SopA from *Salmonella*, SidC from *Legionella pneumophila*, and OspG from *Shigella* partner with UBE2L3 during infection[25]; the anti-microbial role of UBE2L3, if any, also deserves to be explored in the future.

Despite much interest in pro-IL-1β ubiquitylation[16–18,26], the E3 ligases responsible were not known. TRIP12 and AREL1 are ubiquitously expressed, but they have mainly been studied in relation to oncogenesis. For instance, TRIP12, which is known to assemble K29 chains, promotes DNA damage repair[27] and ubiquitin-fusion degradation[28,29], and AREL1 blocks apoptosis by targeting pro-apoptotic proteins[30], and is known to assemble K33- poly-ubiquitin chains[31–33]. Our findings conclusively establish that pro-IL-1β is ubiquitylated and degraded by proteasomes, indicating that IL-1β−driven inflammation could be blocked with PROTACs (proteasome targeting chimeras). PROTACs are heterobifunctional small molecules that leverage the ubiquitin-proteasome system to degrade proteins of interest (POI), including neo-substrates that are not natural substrates of an E3 ligase[34]. PROTACs recruit POIs to a convenient E3 ligase (e.g., CRBN, CRL2[VHL]), resulting in POI ubiquitylation and degradation. Indeed, TRIP12 promotes the complete degradation of the oncogenic transcription factor BRD4 by PROTACs; here the E3 ligase CRL2[VHL] initiates ubiquitylation of BRD4, but TRIP12 adds K29- and K48- chains that enhance degradation[35]. It is tempting to speculate that TRIP12 and AREL1 could be exploited to destabilise their natural substrate pro-IL-1β using a similar approach.

K27-, K29- and K33- poly-ubiquitin chain types have been linked to proteasomal degradation. For instance, like K48- chains, K27- chains facilitate substrate processing by proteasomes[36]. K27- and K29- chains destabilise several innate immune proteins, including NLRP3[37], MDA5 and RIGI[38,39], ULK1[40], STING[41], IRF3 and IRF7[42], among others[43,44].

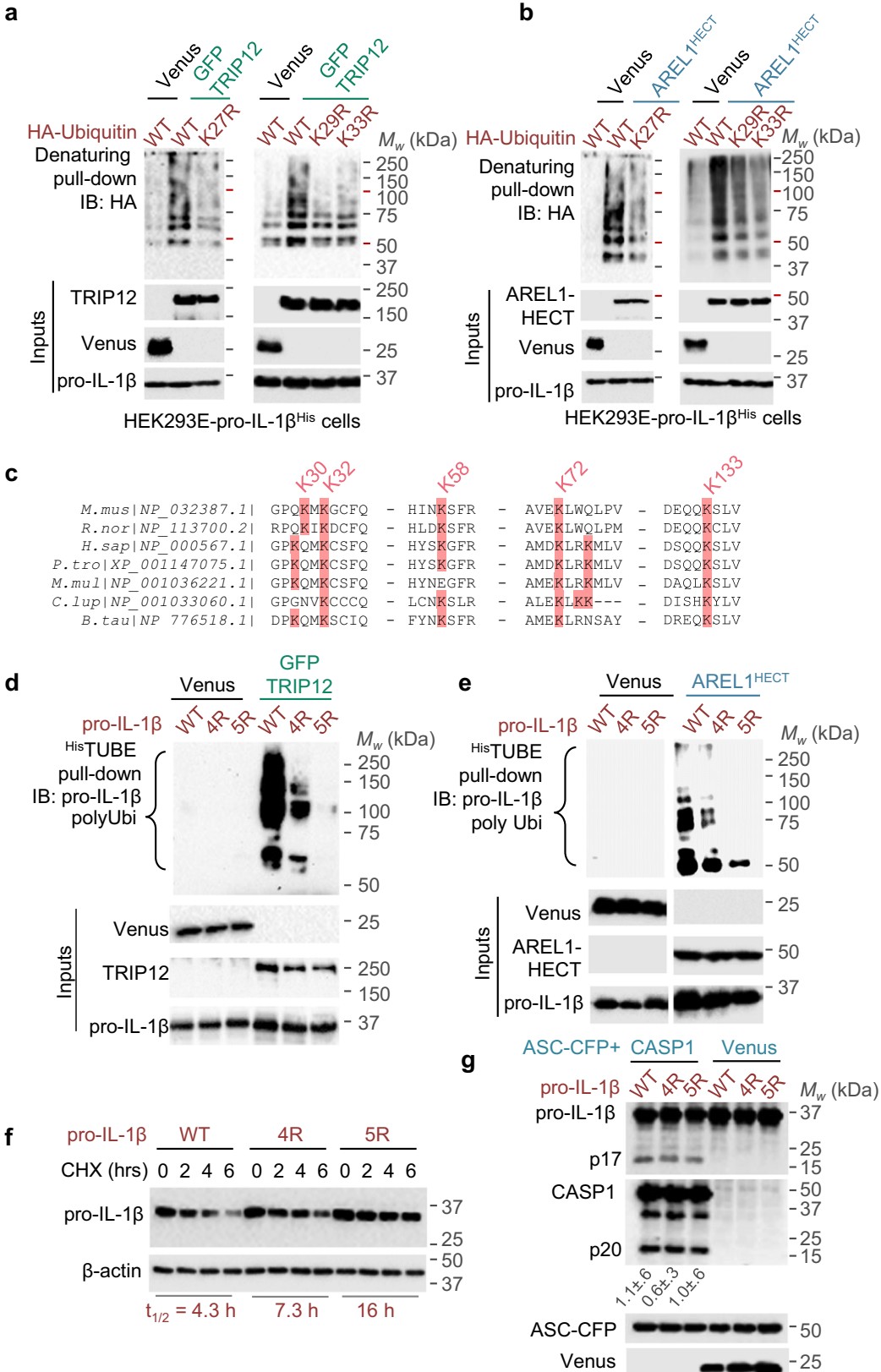

Similarly, K33- chains promote the turnover of RAS GTPases[45], glycine decarboxylase[46], and 3-hydroxy-3-methylglutaryl CoA reductase[46]. UbiCRest (ubiquitin chain restriction) assays with linkage-specific deubiquitylases revealed pro-IL-1β lacks linear ubiquitin but is modified with K11, K48, K63 chains[20]. However, linkage-specific deubiquitylases for K27, K29 and K33 linkages are currently unavailable

commercially, which prevented us and others from testing the presence of these chains on endogenous pro-IL-1β. As TRIP12 and AREL1 did not modify pro-IL-1β with K48- or K11- ubiquitin chains, is plausible that additional E3 ligases participate in pro-IL-1β ubiquitylation with other chain types following the initial ubiquitin chains added by TRIP12-AREL1-UBE2L3. Interestingly, silencing *Trip12* or *Arel1* in

**Fig. 6 | TRIP12 and AREL1 ubiquitylate pro-IL-1β in the N-terminal 'pro' domain. a-b** Representative immunoblots from Ni-NTA pull-downs of pro-IL-1βHis under denaturing conditions to assess the type of ubiquitin chains covalently added on pro-IL-1β by TRIP12 (**a**) or AREL1 (**b**). HEK293E cells stably expressing pro-IL-1β[His] were transfected with HA-tagged wildtype or the indicated K→R mutants of ubiquitin along with GFP-TRIP12 (**a**) or AREL1-HECT[Myc] (**b**) or Venus as negative control. Expression of proteins is shown below (Inputs). Also see Fig. S4B, C for additional blots from the same experiments. **c** Multiple sequence alignment of pro-IL-1β from the indicated organisms showing sequences flanking 'pro' domain lysine residues in the mouse (30, 32, 58 and 72) and human. Sequences around lysine 133 in the mature IL-1β region are also shown. *B. tau* – *Bos taurus*, *C. lup* – *Canis lupus familiaris*, *H. sap* – *Homo sapiens*, *M. mul* – *Macaca mulatta*, *M.mus* – *Mus musculus*, *P. tro* – *Pan troglodytes*, *R. nor* – *Rattus norvegicus*. **d, e** Representative images from TUBE pull-downs assays of wildtype or the indicated K→R variants of pro-IL-1β to assess their ubiquitylation by TRIP12 (**d**) or AREL1 (**e**). HEK293E cells are transfected with plasmids encoding the indicated pro-IL-1β variants (wildtype, 4 R (K30,32,58,72 R) and 5 R (K30,32,58,72,133 R)) and GFP-TRIP12 (**d**) or AREL1-HECT[Myc] (**e**) and Venus as

negative control. Ubiquitylated proteins bound to [His]TUBE were enriched and immunoblotted with anti-IL-1β antibody. Polyubiquitylated pro-IL-1β smears are labelled (polyUbi), and expression of proteins is shown below (Inputs). Images from the same exposure of the same immunoblots are shown in **e** with intervening lanes removed. **f** Representative immunoblots of cycloheximide (CHX)-chase experiments showing increased stability of the indicated variants of pro-IL-1β as compared to wildtype protein. HEK293E cells were transfected with plasmids coding for the indicated pro-IL-1β proteins, and treated with CHX (10 μg.mL⁻¹) for the indicated times 18 h after transfection. Half-life ($t_{1/2}$) calculated from $n = 8$ independent experiments is reported below (also see Fig. S3E). **g** Representative immunoblots showing similar proteolytic maturation of pro-IL-1β variants by caspase-1. HEK293E cells were transfected with plasmids encoding wildtype or mutant pro-IL-1β without or with caspase-1 and ASC-CFP for 18 h, and cell lysates immunoblotted with the indicated antibodies. Ratio of the intensity of mature IL-1β p17 and active caspase-1 p20 bands are indicated (mean ± SD from $n = 3$ independent experiments). Images represent independent experiments (*n*) as follows: **a-b, d-e, g**, $n = 3$; **f**, $n = 8$.

*Ube2l3[ΔMac]* cells did not further impact pro-IL-1β abundance, which indicates that other E2 enzymes cannot compensate for loss of UBE2L3. Future work should clarify whether UBE2L3 can promote pro-IL-1β ubiquitylation via E3 ligases that deploy unusual ubiquitylation mechanisms, such as the RING-cysteine-relay, RING-Zn finger mechanisms or a 'super assembly' of different E3 ligases[47].

Pro-IL-1β ubiquitylation at K133 was conclusively shown to increase its stability in vivo, however, other ubiquitylation were also noted in that study[18]. We have substantially added to these findings by identifying ubiquitylation sites in the 'pro' domain. We focussed on the 'pro' domain after a fortuitous finding while designing the doxycycline-inducible system for the RNAi screen. We found that N-terminal 3xFlag-tag on pro-IL-1β markedly reduced its turnover in macrophages, but C-terminal tagging had no effect (data not shown), which indicated that even small tags interfere with the function of the native N-terminus. We showed the increased abundance of pro-IL-1β through different ways, for example, *Ube2l3*-deletion, silencing of TRIP12 or AREL1 expression, mutation of all four lysine residues in the 'pro' domain (pro-IL-1β-4R) or the additional mutation of K133R (pro-IL-1β-5R), all resulted in increased pro-IL-1β accumulation leading to higher mature IL-1β production. The entire cellular pool of pro-IL-1β is therefore accessible to caspase-1, and the abundance of the pro-IL-1β precursor determines how much mature, inflammatory cytokine is produced. We reason that caspase-1 is not saturated by the levels of pro-IL-1β in wild-type macrophages and the spare proteolytic capacity can contribute to inflammatory outcomes as seen with *Ube2l3*-deletion. Pro-IL-1β ubiquitylation and disposal are therefore critical for homeostasis. The N-terminal 'pro' domain is unique to pro-IL-1α and pro-IL-1β, and K133 is conserved only some of the IL-1 family members (i.e., IL1RA, IL-37 and IL-38). An interesting question to address in the future is the advantage of ubiquitylation in the pro-domain versus the mature domain, and the contribution of TRIP12, AREL1 and UBE2L3 in regulating other IL-1 family members.

Deregulated IL-1β release is linked to cancer and inflammatory conditions that affect major organs, including the vasculature, brain, liver, skin, gut, and joints[1,2]. Currently approved therapies for neutralising IL-1 are biologics such as recombinant IL-1 receptor antagonist (anakinra), anti-IL-1 antibody (canakinumab) or a fusion protein including the ligand-binding regions of the IL-1 receptor 1 and IL1R1 accessory protein (rilonacept)[48]. However, these agents act on the cytokine already in circulation. Pro-IL-1β released from necrotic/pyroptotic cells might also be processed extracellularly[1,2]. Targeting intracellular pro-IL-1β, for example, via the proteasome, could potentially be more effective as an anti-inflammatory therapeutic intervention. Our identification of the ubiquitylation mechanisms provide a foundation for such work in the future.

## Methods

### Ethics statement
All work with mice was performed in accordance with the Animals (Scientific Procedures) Act 1986 and was approved by the local ethics review committee at Imperial College London (PILs P9A000710 and PP9741955 holder ARS). Both male and female mice (age 2–4 months) were used for harvesting bone marrow-derived macrophages and in vivo experiments. Mice were euthanised by a Schedule 1 method (cervical dislocation followed by exsanguination as confirmation of death).

### Reagents
Details on antibodies used, their sources, dilutions and validation are provided in Supplementary Table 1; all other reagents and their sources, including plasmids, siRNA and primer sequences used in this study are listed in Supplementary Table 2.

### Generation of *Ube2l3[ΔMac]* mice
Briefly, the strategy to generate *Ube2l3*-deficient mice involved flanking exon 1 of *Ube2l3* (Gene ID 22196 on chromosome 16), which contains the translation start site, with two LoxP sites to enable its deletion by Cre-driven recombination. *Ube2l3[fx/+]* mice were generated by Taconic Biosciences on an ES cell line in the C57BL/6NTac background. These mice were crossed with mice expressing a modified oestrogen receptor fused to the Cre recombinase (icre/Esr1*) under the promoter of Csf1r (also called CD115 or MCSF1R)[12]. This results in deletion of *Ube2l3* exon 1 in macrophages and Csf1r-positive myeloid cell populations upon administration of tamoxifen to mice. Csf1R-cre/Esr1* mice backcrossed 9 generations to C57BL/6 background (B6-Tg(Csf1r-cre/Esr1)1Jwp GA) were a kind gift from Jeffery W Pollard, University of Edinburgh[12]. Mice were bred at Imperial College to generate *Ube2l3[fx/fx] cre/Esr1* mice, sperm from which were used to rederive them (MRC Transgenic Facility) into the local specific pathogen-free (SPF) facility. Subsequent crosses involved homozygous *Ube2l3[fx/fx]* mice mated with similar homozygous mice carrying a single copy of Csf1r-icre/Esr1*. For convenience, mice with conditional deletion of *Ube2l3* in Csf1r-positive cells (Csf1r is highly expressed in macrophages and monocytes, with lower expression in granulocytes and lymphocytes[12,49]) are called *Ube2l3[ΔMac]* here. All mice were genotyped at weaning (21 day old) using ear biopsies. Mice of unwanted genotypes were humanely euthanised. To induce recombination, mice were orally gavaged on 3 consecutive days with 80 mg.kg⁻¹ body weight tamoxifen prepared in corn oil. Two days after the last tamoxifen treatment, *Ube2l3* deletion was confirmed by three independent methods: PCR analysis on genomic DNA from peritoneal cells using specific primers (designed by Taconic) to verify homologous recombination;

flow cytometry analysis of peritoneal cells using antibodies against CD115 and UBE2L3; western blot analysis on adherent peritoneal cells using UBE2L3-specific antibody. Subsequently, all experiments were performed 48 h after the third tamoxifen dose.

Mice were housed and bred in dedicated animal facilities of Imperial College London (12 h light/dark cycle; 22 +/-2 °C; 30 to 40% humidity). Mice were housed IVC cages with corn cob bedding and enrichments, including nesting material, refuges, and gnawing sticks. Mice were fed with RM1(E) rodent diet (SDS Diet, Dietex, UK) and water *ad libitum*.

## Flow cytometry
Cells ($1 \times 10^6$) collected from the peritoneal lavage were washed in flow cytometry staining buffer and incubated with 1:100 dilution of anti-mouse CD16/32 (Fc block) antibody for 10 min at room temperature. Cell surface marker staining was performed using 1:100 dilution of respective antibodies (e.g., anti-mouse F4/80-APC, anti-mouse Ly6G-APC, or anti-mouse CD115 (Csf1r)-APC) in the dark for 30 min at 4 °C in staining buffer. Cells were washed thrice with staining buffer followed by fixation with IC fixation buffer for 15 min. For intracellular antigen staining, cells were washed thrice with permeabilization buffer and incubated with permeabilization buffer for 10 min at 4 °C. UBE2L3 was stained using anti-mouse Ube2l3-FITC antibody at 1:100 dilution in dark for 30 min at 4 °C in permeabilization buffer, followed by three washes in staining buffer. Total neutrophils were estimated by staining cells with anti-mouse Cd11b-FITC and anti-mouse Ly-6G-APC. Data were collected on a BD FACSCalibur™ and analysed with Cyflogic and FlowJo software. Gating strategies are shown in Fig. S5.

## LPS endotoxic shock models in mice
Mice were treated with 2.5 mL of 3 % sterile thioglycolate intraperitoneally to induce peritoneal macrophages. Two days after thioglycolate injections, *Ube2l3* deletion was induced in mice by oral administration of tamoxifen as described above. For the high-dose model, mice were given 30 mg.kg$^{-1}$ body weight of LPS prepared in sterile PBS for 3 h intraperitoneally. Disease severity was measured as the sum of four parameters: coat fur, mobility, bulging of eyes and posture; scores ranged from 0 to 4, wherein 0 scores for a healthy mouse. Scoring was done by a trained professional blinded to the treatment and genotype of mice. Mice were euthanised and blood was collected for serum cytokine analyses.

The low-dose LPS model involved thioglycolate and tamoxifen treatments as above. Age and sex-matched mice weighing 21-22 g were given LPS (5 µg in sterile PBS) intraperitoneally for 3 h followed by an intraperitoneal injection of 50 µmol of ATP (in sterile PBS). Ten minutes after ATP treatment, mice were euthanised, blood and peritoneal lavage were collected for cytokine analyses. Peritoneal lavage was collected by injecting 3 mL sterile ice-cold Dulbecco's modified PBS containing 0.1 % BSA.

For *Ube2l3* depletion post inflammasome activation, in vivo peritonitis was induced in thioglycolate-treated *Ube2l3*fx/fx mice as described above, and peritoneal lavage was collected, and $1 \times 10^6$ cells were stained for flow cytometry analysis as described below and supernatant was assayed for cytokines using manufacturer's protocol.

## MSU crystal-induced peritonitis
MSU-induced peritonitis was performed as described previously[4,50]. In brief, following tamoxifen regimen as above, adult mice were injected intra-peritoneally with 1 mg MSU crystals resuspended in 0.2 mL PBS. After 6 h, mice were euthanized and injected with 1 mL PBS into the peritoneum. Peritoneal lavage was collected, and $1 \times 10^6$ cells were stained for flow cytometry analysis and supernatant was assayed for IL-1β using manufacturer's protocol.

## Bone-marrow derived macrophages
Bone marrow-derived macrophage cells (BMDMs) were prepared as described before[6]. Briefly, femur, tibia, fibula were excised from mice, cleaned in 70 % ethanol and DMEM containing gentamicin (50 µg.mL$^{-1}$), and cells in the marrow flushed out with 5 mL DMEM. Single-cell suspensions were generated by vigorous pipetting and used for differentiation into macrophages or stored in 90% heat-inactivated foetal bovine serum (HI-FBS) containing 10 % DMSO for future use. Differentiation was carried out in non-cell culture-treated (bacterial) 10 cm petri plates in medium containing 20 % conditioned medium from L929 fibroblast culture. Macrophages were used for experiments after day 6 of differentiation. BMDMs were immortalized using J2CRE virus as described before to generate immortalized BMDMs (iBMDMs) and reduce the use of mice[6,51]. iBMDMs were maintained in Dulbecco's modified Eagle's medium (DMEM) supplemented with 10 % HI-FBS, 100 U.mL$^{-1}$ penicillin and streptomycin, and 20 % L929 conditioned medium.

## Cell culture
Primary BMDMs and iBMDMs were grown in complete DMEM (high-glucose DMEM, 10 % HI-FBS, 50 U.ml$^{-1}$ penicillin and 50 µg.ml$^{-1}$ streptomycin) supplemented with 20 % L929 conditioned-medium. Puromycin (6 µg.ml$^{-1}$), doxycycline (500 ng.mL$^{-1}$) were added when needed. Human embryonic kidney 293E (HEK293E) and L929 cells were grown in complete DMEM medium. Cells were routinely tested and found negative for mycoplasma contamination.

## Treatments of cells in vitro
To induce *Ube2l3* knock-out in primary or immortalized *Ube2l3*fx/fx and *Ube2l3*fx/fx iCre+ BMDMs, cells in 10 cm dishes were treated with 2 µM of 4-hydroxytamoxifen (hTam) for 48 h, followed by trypsinisation to harvest cells which were then plated for experiments.

For cycloheximide (CHX)-chase experiments, hTam-treated or siRNA transfected iBMDMs were plated in 48 well plate at a seeding density of $1.5 \times 10^5$ cells per well. Cells were treated with 250 ng.mL$^{-1}$ of LPS for 14 h followed by treatment with 10 µg.mL$^{-1}$ of CHX for times as indicated. Inhibition of proteasomes was achieved by treating cells with 10 µM of MG-132.

To determine the effect of various chemical inhibitors of protein degradation pathways on pro-IL-1β abundance, cells were primed with LPS (250 ng.mL$^{-1}$) or doxycycline (500 ng.mL$^{-1}$) and followed by treatment with MG-132 (10-25 µM), Bafilomycin A (BafA; 20 nM), BAY 11-7082 (50 µg.mL$^{-1}$), MLN4924 (3 µM) and Heclin (100 µM) at times as indicated in figure legends.

To determine the levels of mature IL-1β in primary and immortalized BMDMs, cells were treated with 250 ng.mL$^{-1}$ LPS for 3 h, followed by 50 µM of nigericin for 1 h. For western blots, samples were treated and processed as described before[6]. Secreted IL-1β was measured using ELISA and western blot analysis of supernatant from cells. Cell cytotoxicity was measured using propidium iodide (PI) dye uptake assay[52]. Cells lysed with Triton-x100 was used to obtain the percentage of the PI uptake in treated cells and untreated cells were used as baseline controls.

## Immunoblotting
Samples for immunoblotting were prepared in RIPA buffer (60 mM Tris pH 8.0, 150 mM NaCl, 1 % NP-40, 0.5 % sodium deoxycholate, 1 mM EDTA) supplemented with 1x protease inhibitor and 1 mM PMSF, and then mixed with Laemmli loading buffer (50 mM Tris pH 6.8, 2 % (w/v) SDS, 10 % (v/v) glycerol, 0.01 % (w/v) bromophenol blue, 0.05 % (w/v) 2-mercaptoethanol)[6,52,53]. Extracts were separated by SDS-PAGE using Tris-Glycine buffer systems and transferred to PVDF membranes. Blots were incubated with the indicated primary and secondary antibodies. Immunoblots were developed with Clarity Western ECL for cell lysates and ECL Prime for supernatant samples. In all images, molecular

weights markers are marked in kDa units based on the migration of Precision Plus Protein Dual Color Standards.

## siRNA screen and analyses

To identify the E3 ligases involved in the ubiquitylation of pro-IL-1β, we used iBMDMs stably expressing pro-IL-1β under doxycycline-inducible promoter coded in the pLTREK-2P-mIL-1β$^{Strep}$ vector (derived from a vector described previously[53]). A cherry-picked library of SMART Pool siRNA for mouse HECT and RBR ubiquitin E3 ligases and non-targeting controls was obtained from Dharmacon™ where genes were randomly assigned to wells in a 96-well plate, including empty wells, non-targeting controls, *rtta3* siRNA. The screen was performed three independent times with technical duplicates within each repeat. Experimenters and analysts were blinded to gene names until data analyses.

Briefly, iBMDMs pLTREK-2P-mIL-1β$^{Strep}$ stable cells were seeded in a 96-well plate at a density of $3 \times 10^4$ cells/well 24 h before siRNA transfection. Transfections were performed with TransIT-X2 Dynamic Delivery System (25 nM final concentration of siRNA in wells) for 72 h according to manufacturer's procedures. Cell culture medium was changed 48 h after transfection, and treatments were performed the following day. Cells were treated with doxycycline (500 ng.mL$^{-1}$) for 15 h, washed twice and chased for 3 h for turnover (total 18 h treatment). "High" controls (control siRNA + 3 h MG-132 treatment (to block proteasomal turnover) or 15 h doxycycline treatment (no 'chase')) and "low" controls (no doxycycline treatment, *rtta3* gene silencing, 15 h doxycycline treatment + 3 h 'chase') were done in wells randomly placed in the 96-well plate. After treatment, cells were lysed with PBS, 0.1 % Tween 20, protease inhibitors and 1 % BSA followed by 2 freeze-thaw cycles. Pro-IL-1β levels were measured by ELISA (88-8014-22; Thermo Fisher), and cell death was determined in the supernatant using the Promega LDH release assay kit. Each siRNA screen attempt included technical duplicate transfections, and means from three independent repeats were analysed in Microsoft Excel and R, and plotted in R. Robust z score (z*), robust strictly standardized mean difference (SSMD*) and P value cut-offs were calculated following published methods[54]. 'Robust' z scores use medians and median of the absolute deviation (MAD) and are less affected by outliers than mean and standard deviation. Briefly, log-transformed value of pro-IL-1β in the well that received non-targeting control siRNA ('low control') was subtracted from log-transformed values for all other wells followed by the calculation of z* scores for all siRNA target wells within each biological repeat. SSMD* scores were calculated from z* from three independently repeated screens. Mean z* from three experiments and SSMD* are plotted in Fig. 4D, E. P values were calculated following linear mixed-effects analyses (random intercepts allowed for each experiment) and false discovery rate (FDR)-adjustment for comparisons for each siRNA against 'low control', i.e., non-targeting control siRNA-treated wells incubated up to 18 h for pro-IL-1β turnover. Genes whose silencing led to increased pro-IL-1β abundance were expected to have a low P value and relatively high z* and SSMD* scores.

## Quantitative RT-PCR

RNA extraction was performed using the RNeasy mini kit according to the manufacturer's protocol. Reverse transcription used 2 μg of purified RNA and High-capacity cDNA Reverse Transcription kit. Quantitative PCR was performed using SsoAdvanced Universal SYBR green supermix on a StepOnePlus Real-Time PCR system. Reactions were performed in duplicate, including negative control lacking cDNA or primer. Data are expressed as $2^{-\Delta CT}$ values normalised to *Gapdh*.

## Enzyme-linked immunosorbent Assays.

Cytokines were detected by ELISA from serum and peritoneal lavage samples of mice, and the supernatant of treated cells or cell lysates. The following ELISA kits were used to determine cytokine concentration according to the manufacturer's instructions: mouse pro-IL-1β (88-8014-22; Thermo Fisher), mouse IL-1β (88-7013-88; Thermo Fisher), mouse TNFα (88-7324; Thermo Fisher), mouse IL-6 (88-7064; Thermo Fisher).

## Recombinant $^{His}$TUBE expression and purification

BL21-CodonPlus (DE3)-RIPL cells were transformed with pPRO_Ex-HIST-TEV_6xTR-TUBE, and protein expression was induced with 100 μM IPTG at 18 °C for 18 h. Cells were collected, washed in ice-cold PBS and lysed by sonication in lysis buffer (50 mM Tris-HCl pH 7.0, 300 mM NaCl, 15 mM imidazole, 5 mM 2-ME and 1 mM PMSF). Initial Ni$^{2+}$-NTA purification was carried out using a HisTrap HP column (5 mL column, GE Healthcare) on an ÄKTA Start, with all buffers being ice-cold and filtered through a 0.2 μm membrane. Unbound material was washed with 10 column volumes (CV) of wash buffer (50 mM Tris-HCl pH 7.0, 100 mM NaCl, 15 mM imidazole and 5 mM 2-ME) followed by a 10 CV gradient elution over 50 fractions in elution buffer (50 mM Tris-HCl pH 7.0, 100 mM NaCl, 200 mM imidazole, 5 mM 2-ME). Fractions containing $^{His}$TUBE (theoretical pI 3.69) were diluted in no-salt buffer (50 mM Tris-HCl pH 7.0, 5 mM 2-ME) to a final NaCl concentration of 10 mM. The sample was then loaded onto a HiTrap Q HP anion exchange column (5 mL, GE Healthcare), followed by washing unbound material using 10 CV of AEX wash buffer (50 mM Tris-HCl pH 7.0, 10 mM NaCl, 5 mM 2-ME) and elution over a 10 CV gradient of AEX elution buffer (50 mM Tris-HCl pH 7.0, 1 M NaCl, 5 mM 2-ME). Fractions containing $^{His}$TUBE were pooled before concentration and desalting into storage buffer (50 mM Tris-HCl pH 7.0, 100 mM NaCl, 5% glycerol, 5 mM 2-ME) through two centrifugation steps using a Vivaspin 20 MWCO 5 kDa Centrifugal Filter Column. The protein concentration was estimated using the Bradford assay, and recombinant $^{His}$TUBE was snap frozen in storage buffer and stored at -80 °C until use, with ~20 μg protein used per pull-down.

## Tandem ubiquitin-binding entities (TUBE) pull-down

HEK293E cells ($4 \times 10^5$) were plated in 6-well plates and transfected with either wildtype or mutant mouse IL-1β along with either plasmid encoding Arel1-Hect$^{MYC}$ (HECT domain of Arel1), AcGFP-Trip12 (referred to as GFP-Trip12 in the text) or mVenus (referred to as Venus in the text) as the negative control. One day post-transfection, cells were treated with 10 μM MG132 for 3 h, washed with ice-cold PBS and harvested in 500 μL of lysis buffer (25 mM Tris pH 7.0, 150 mM NaCl, 1 % NP-40, 20 mM imidazole, 10 mM freshly prepared N-ethyl maleimide (NEM)) containing 1 mM PMSF, and protease and phosphatase inhibitor cocktails. Harvested cells were incubated for 20 minutes on an end-over rocker at 4 °C, followed by 3 cycles of 10 seconds each of sonication at 30 % amplitude (Sonics, VibraCell). Lysates were centrifuged (14,000 x*g*, 30 min, 4 °C) and supernatants incubated with $^{His}$TUBE overnight at 4 °C on an end-over rocker. The $^{His}$TUBE protein along with ubiquitylated proteins was pull-down using Ni-NTA magnetic beads for 1 h at 4 °C, followed by four washes in lysis buffer, and elution of bound proteins in reducing and denaturing Laemmli loading buffer for western blot analysis.

## HA-ubiquitin pull-down assays

HEK293E cells with stable expression of hexa-histidine-tagged mouse pro-IL-1β were generated using retroviral transduction of HEK293E cells with pMX-CMV-mIL-1β-His plasmid, followed by the selection of transduced cells with 2 μg/mL of puromycin. His-tagged mouse IL-1β expression in selected cells was confirmed using western blot analysis. HEK293E-mIL-1β$^{His}$ cells ($4 \times 10^5$) were plated in 6-well plates and transfected with either plasmid encoding HA-tagged wildtype

ubiquitin or HA-tagged mutant ubiquitin with either plasmid encoding Arel1-HECT$^{Myc}$, AcGFP-Trip12 or mVenus as negative control. At 24 h post-transfection, cells were treated with 10 μM of MG132 for 3 h, washed with ice-cold PBS and harvested in 500 μL of denaturing lysis buffer (25 mM Tris pH 7.0, 300 mM NaCl, 1 % NP-40, 15 mM imidazole, 8 M Urea, 10 mM NEM) containing 1 mM PMSF, and protease and phosphatase inhibitor cocktails. Harvested cells were incubated 20 minutes on an end-over rocker at 4 °C, followed by 3 cycles of 10 seconds each of sonication at 30% amplitude (Sonics, VibraCell). Lysates were centrifuged (14,000 x*g*, 30 min, 4 °C) and supernatants incubated with magnetic Ni-NTA beads overnight at 4 °C on an end-over rocker. Following four washes in denaturing lysis buffer, bound proteins were eluted using reducing and denaturing Laemmli loading buffer for western blot analysis using anti-HA antibody.

## Pro-IL-1β turnover and caspase-1 cleavage assays in HEK293E cells

For wildtype, 4 R and 5 R pro-IL-1β$^{Strep}$ turnover assays, HEK293E cells (4 ×10$^5$) were plated in 6-well plates and transfected with plasmid encoding the corresponding variant. One day post-transfection, cells were harvested with trypsin and plated in five wells of a 24-well plate and left overnight. Cells were treated with cycloheximide (10 μg.mL$^{-1}$) for times as indicated in figures and total cell lysates were prepared in RIPA buffer as above for immunoblot analysis. For caspase-1 cleavage, HEK293E cells (1 ×10$^5$) were plated in 24-well plates and transfected with plasmid encoding either wildtype or mutant mouse pro-IL-1β, mASC-CFP and either mouse caspase-1 or mVenus as negative control. Cell lysates were prepared 18 h post-transfection for immunoblot analysis.

## Co-Immunoprecipitation

HEK293E cells (4 ×10$^5$) were plated in 6 well plates and transfected with either wildtype mouse IL-1β or UBE2L3$^{His}$ along with either plasmid encoding mVenus-tagged Arel1, AcGFP-tagged Trip12 or plasmid encoding mVenus as negative control. For Co-immunoprecipitation of Arel1 and Trip12, cells were transfected with plasmid encoding AcGFP-tagged Trip12 along with either mVenus-Arel1$^{Myc}$ or mVenus$^{Myc}$ as negative control. One day post-transfection, cells were treated with 10 μM MG132 for 3 h, washed with ice-cold PBS and harvested in RIPA buffer containing 1 mM PMSF, and protease and phosphatase inhibitor cocktails. Harvested cells were lysed on an end-over rocker at 4 °C for 20 minutes, followed by 3 cycles of 10 seconds each of sonication at 30 % amplitude (Sonics, VibraCell). Lysates were centrifuged (14,000 x*g*, 30 min, 4 °C), and supernatants used for immunoprecipitation with 1 μg of anti-GFP antibody for mouse IL-1β and UBE2L3$^{His}$ IP with Arel1 or Trip12, or 1 μg of anti-Myc antibody for Arel1 and Trip12 co-IP. Samples were incubated on an end-over rocker for 18 h at 4 °C, followed by 2 h incubation with 40 μl of magnetic protein G-Sepharose slurry to capture antigen-antibody complexes. Beads were washed thrice with RIPA lysis buffer; each wash was carried out for 10 min on an end-over rocker at 4 °C. Protein complexes bound to the beads were eluted with 50 μl of 2× Laemmli loading buffer and analysed by western blotting.

## Molecular cloning

Molecular cloning was performed using one-step sequence and ligation-independent cloning (SLIC)[55], and all constructs were confirmed by DNA sequencing (GATC Biotech or Genewiz). Doxycycline-inducible expression used an in-house one-step lentiviral plasmid like pLTREK3-TEV-T2-GFP used previously[53]. Briefly, mouse IL-1β cDNA was cloned downstream of the Tet-inducible promoter from pRetroXTight (Takara), with a Kozac sequence and C-terminal 2x StrepTag II (WSHPQFEKGGGSGGGSGGGSWSHPQFEK) tags. The TetOn transcription factor rtTA3 gene, a self-cleaving P2A peptide (GSGATNFSLLKQAGDVEENPGP) and puromycin-resistance gene were cloned downstream of the PGK promoter to obtain pLTREK-2P-

mIL-1βStrep. pMxCMV-YFP and pMxCMV-YFP-UBE2L3 plasmids were described before[6]. Similarly, pro-IL-1β constructs with 2x C-terminal StrepTag II tags were cloned into pMX-CMV plasmid for constitutive expression.

Mouse AREL1 cDNA obtained from Origene (MR210828) and the different constructs (HECT-domain: aa 436-823) were cloned with two C-terminal 2x-Myc tags or N-terminal mVenus and C-terminal 2x-Myc tags into pMX-CMV-mVenus-2Myc via PCR. A T424I polymorphism in the commercial cDNA was corrected to generate I424T AREL1 for subsequent cloning. Human TRIP12 plasmid (pAcGFP-TRIP12) was a kind gift from Jiri Lucas (University of Copenhagen)[56].

Site-directed mutagenesis used single oligonucleotide mutagenesis-based linear PCR as described previously[57]. In some cases, two mutagenic primers with ~18-bp identical ends (for SLIC cloning) were used to amplify the entire plasmid by PCR with the hot-start KOD polymerase followed by SLIC[55]. This used KOD polymerase buffer containing 2.25 mM MgSO$_4$ and 2.5 % DMS). Cycling conditions used two-stage programmes with first 10 cycles used annealing temperature of the primers, followed by 10 cycles at which annealing temperature increased by 0.5 °C increase per cycle, followed by 10 cycles of 2-step PCR consisting of denaturation (94 °C) and amplification (70 °C) without an annealing step. This method was used to generate the following mutations in pMXCMV-pro-IL-1β$^{Strep}$: K30R, K32R, K58R, K72R, and K133R, and their combinations to generate pMXCMV-pro-IL-1β$^{Strep}$-4R (K30R, K32R, K58R, K72R) and pMXCMV-pro-IL-1β$^{Strep}$-5R (K30R, K32R, K58R, K72R, K133R).

TR-TUBE (4 tandem repeats of the trypsin-resistant UBQLN1 UBA domain tagged with N-terminal hexahisitine tag) from pRSET-6xTR-TUBE (Addgene, #110313, kind gift from Yasushi Saeki[20],) was digested by BamHI and SalI enzymes and cloned into pPRO-Ex-HT-A vector to obtain pPRO_Ex-HIST-TEV_6xTR-TUBE.

## Retroviral and lentiviral transduction

For retro- (e.g., pMX-CMV plasmids) and lenti- (e.g., pLTREK plasmids) viral transduction, plasmids were transfected with Lipofectamine 2000 into HEK293E cells plated in 12-well plates for packaging pseudotyped virus-like particles. A total of 1.5 ug DNA per well consisting of target plasmid: pCMV-MMLV GagPol (for retroviral vectors): pCMV-VSV-G at a ratio of 5:4:1 or target plasmid: pHIV GagPol (for lentiviral vectors): pCMV-VSV-G at a ratio of 4:3:2, was used. Culture supernatants were collected after 48 h after addition of sterile cell culture-grade Hepes buffer (pH 7.4, final concentration 10 mM) and filter-sterilised (0.45 μm non-PVDF 13 mm sterile filters) for transduction. Target cells plated in one well of a 12-well plate in 1 mL medium received ~0.3-0.4 mL of packaged virus for 48 h, followed by removal of medium and addition of fresh medium containing the appropriate selection antibiotic.

## Statistical analysis and data plotting

Experiments with mice were planned as randomized blocks with 2-3 animals per genotype per experiment, and experiments were repeated independently 2-5 times. Mice were genotyped for the presence of *Ube2l3*$^{fx/fx}$ without or with Csf1R-Cre-ERT2 gene and randomly assigned to tamoxifen or corn oil groups, and were genotyped again after sacrifice to independently verify the presence or absence of the *Cre* gene. For high-dose LPS model, disease activity scores were determined by an experimenter blinded to genotypes and treatments. Investigators were not blinded to treatments in other experiments. Age- and sex-matched male and female mice of age between 8-12 weeks were used across experiments. Data from all mice in all experiments were pooled and analysed using mixed effects models with 'Experiment' as the blocking factor (variable intercepts) using lme4[58], lmerTest[59], and emmeans[60] packages as implemented in the grafify[61] package in R.

In vitro experiments, such as ELISA, LDH-release, PI-uptake, real-time qRT-PCR, and others were set up as two to three technical replicate wells, values from which were averaged to obtain a mean for that experiment. Experiments were independently repeated with fresh source/passage of cells on different days and means from different experiments (denoted by n in figure legends) were analysed statistically using mixed effects models with 'Experiment' as random factor (variable intercept models). When performing the siRNA screen, experimenters were blinded to gene names in wells until the analyses of independent repeats.

When linear mixed effects model diagnostics (e.g., ggResidpanel[62] and performance[63] packages in R) revealed major deviation of residuals from the normal distribution, data transformations (e.g., logarithms, logit) were used. False discovery rate (FDR; $Q = 5$ %) was used to correct for multiple comparisons as implemented in emmeans in R or GraphPad Prism. FDR-adjusted $P$ values > 0.05 were considered non-discoveries or non-significant (ns). Non-linear least square fit predictions and their 95 % confidence intervals used investr package[64]. Graphs were generated in R using grafify[61] and ggplot2[65] packages, using dotplot, boxplot and violin geometries. Dotplots (default binwidth 1/30 of Y axis) are shown with data distribution depicted by a box (showing $1^{st}$ and $3^{rd}$ quartiles), with horizontal line at median and whiskers ( ± 1.5 interquartile range (IQR)), and violins (trimmed to minimum and maximum data values). Where used, error bars are described in figure captions.

### Reporting summary

Further information on research design is available in the Nature Portfolio Reporting Summary linked to this article.

## Data availability

Source data are provided in the Source Data file. Further information and requests for resources and reagents should be directed to Avinash Shenoy (a.shenoy@imperial.ac.uk). Source data are provided with this paper.

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

## Acknowledgements

Authors would like to acknowledge funding from the MRC (MR/P022138/1, MR/T00004X/1, and MR/V030930/1 to ARS). We also acknowledge the MRC-funded High-throughput Single-Cell Analysis facility (HTSCAF; MR/P028225/1) at the MRC CMBI, MRC Transgenic Facility at Imperial College London for rederiving mice, and Jessica Rowley (Manager, flow cytometry) for help with flow cytometry. CM would like to acknowledge the doctoral training programme award to MRC CMBI (MR/R502376/1). We thank Jeffery W Pollard for sharing Tg(Csf1r-cre/Esr1)1Jwp GA) mice on C57/BL6 background, and Jiri Lukas and Hiroshi Ashida for plasmids. The authors thank Sandhya Visweswariah and Gad Frankel for discussions, Teresa Thurston and David Holden for critical comments on a draft of the manuscript.

## Author contributions

Investigation: A.C.-P., V.M., C.M., T.M., I.G., A.D., G.D., M.J.G.E., L.M., M.M., S.W., D.B., R.M., A.R.S.; Validation: A.C.-P., V.M., G.D., M.M., S.W.; Visualisation: V.M., C.M., A.C.-P., A.R.S.; Writing – original draft: A.R.S.; Writing – review and editing: A.R.S., V.M., A.C.-P., D.B.; Supervision: A.R.S., V.M., A.C.-P., D.B., Software: A.R.S.; Formal analysis: V.M., A.C.-P., C.M., R.M., A.D., A.R.S.; Funding acquisition: A.R.S.; Conceptualisation: A.R.S.

## Competing interests

The authors declare no competing interests.
