## [Peer Review File · Nature Communications]

IL-1 β turnover by the UBE2L3 ubiquitin conjugating enzyme and HECT E3 ligases limits inflammationREVIEWER COMMENTS

Reviewer #1 (Remarks to the Author):

The manuscript by Mishra, Crespo-Puig et al. follows up on this group's published finding that the E2 ligase UbcH7 regulates pro-IL-1b stability. In the current work, the authors extend that Cell Reports paper to generate a Tamoxifen-inducible, myeloid-specific UbcH7 knockout mouse. They show that that mouse has increased IL-1b levels and release upon inflammasome activation with an increased half-life of pro-IL-1b. They further develop an siRNA screening assay to determine the E3 ubiquitin ligase(s) responsible for IL-1b ubiquitination. Using that system, they find that Arel1 and Trip12 mediate IL-1b ubiquitination. The manuscript is logical, very well-written and really a joy to read. I think it will be of great interest to the innate immune and inflammasome communities. There are some points and experimentation that need clarification and/or extension, however. My specific comments are as follow:

1. In Figure 1G, it isn't clear why UbcH7 mRNA levels fall in response to LPS + ATP. This could simply be an artifact of the cell undergoing pyroptosis. This needs to be examined further, perhaps with a GasderminD inhibitor, LPS and ATP alone and/or Gasdermin D knockout cells.

2. It isn't clear from the manuscript if the UbcH7-knockout myeloid mice have an inflammatory phenotype or inflammatory pathology. What does a necropsy show? Along these same lines, the authors are injecting LPS into these mice. Did the authors do an LPS-sepsis study? Presumably the UbcH7 mice would show lethality at a lower dose of LPS?

3. Figure 3E would be more convincing with a time course of IL-1b release and cleavage as well as a time course of Gasdermin D cleavage. It isn't clear until Figure 5I that pyroptosis (and presumably Gasdermin D cleavage) is unaffected in the UbcH7-null mice.

4. It should be at least mentioned in Figure 4A that BAY11-7082 is an IKK inhibitor. It won't matter here as they aren't looking at endogenous pro-IL-1b, but NF- κ B inhibition could be affecting their system and it should at least be mentioned in the text.

5. Not a critique, but I find the siRNA screen to be elegant and well-thought out with very good validation. It wasn't clear to me as a reader, however, how the authors settled on HECT/RBR E3 ligases. I'm assuming that these are the ones known to use UbcH7, but a sentence or two in the manuscript formally explaining this would be useful.

6. In Figure 5 and 6, it isn't clear the relative contributions of TRIP12 and Arel1 to the total phenotype of the UbcH7-null cells. That is, what happens when you compare levels of IL-1b in when you use siRNA to knockdown UbcH7, Trip12 and Arel1 in the same system? Do these two E3 ligases mediate all of the UbcH7 effect? Are they synergistic or parallel?

7. The authors should be commended for using denaturing conditions in their ubiquitination experiments in Figure 6. Not doing this is a common cause of ubiquitination artifacts and this attention to detail is quite nice to see.

In summary, I find the work as a whole to be well done and extremely interesting. I think attention to the concerns raised above would improve the manuscript, but overall, I quite enjoyed reading it.

Reviewer #2 (Remarks to the Author):

In this study, it is reported that deletion of Ube2l3 in mice markedly reduces pro-IL-1 β turnover in macrophages, leading to excessive mature IL-1 β production, neutrophilic inflammation and disease symptoms following inflammasome activation. The authors use a family-wide siRNA screen identified two ubiquitin ligases, TRIP12 and AREL1, which add K27-, K29- and K33- poly-ubiquitin

chains on lysine residues in the 'pro' domain and destabilise pro-IL-1 β . Mutation of ubiquitylation sites increased pro-IL-1 β stability, but did not affect proteolysis by caspase-1. The extent of mature IL-1 β production is determined by precursor abundance, and UBE2L3, TRIP12 and AREL1 limit inflammation by shrinking the cellular pool of pro-IL-1 β . This study uncovered fundamental processes governing IL-1 β homeostasis and provided molecular insights that could be exploited to mitigate its adverse actions in disease. However, this article still has many shortcomings to be improved.

Major points:

1. The article called "The Atypical Ubiquitin E2 Conjugase UBE2L3 Is an Indirect Caspase-1 Target and Controls IL-1 β Secretion by Inflammasomes" (DOI: 10.1016/j.celrep.2017.01.015) has already revealed that UBE2L3 acts post-translationally to promote K48-ubiquitylation and turnover of pro-IL-1 β and dampen mature-IL-1 β production and UBE2L3 depletion increases pro-IL-1 β levels and mature-IL-1 β secretion by inflammasomes, so much of the work in this article seems to be just repetition and further corroborate this conclusion.
2. Mice were given PBS, LPS (5 μ g, 3 h) or LPS+ATP (5 μ g, 3 h + 50 μ mol, 10 min) intraperitoneally, it seems that the dose should be based on the weight of the mouse, rather than simply how much reagent to give a mouse.
3. On line 110, the paper says that UBE2L3 could act as a negative regulator of IL-1 β production in vivo, but the above data alone does not seem to lead to such a conclusion.
4. In Figure S1, the result of PCR-based screening for genotyping WT (Ube2l3^{fx/fx}) and KO (Ube2l3 Δ Mac) primary BMDMs given 4-hydroxytamoxifen (2 μ M) for 48 h is inconsistent with Oligonucleotide primer binding sites indicated.
5. In Figure 2, the data of three established models of inflammasome—dependent inflammation and quantified IL-1 β production and inflammatory parameters has a surprisingly large margin of error in each experiment group, what do you think caused this and how to improve this?
6. On line 175-176, the author said that these results from knockout mice unequivocally establish a crucial role for UBE2L3 in acting 'upstream' of inflammasomes by reducing pro-IL-1 β abundance, this is questionable and the results only indicate that deletion of Ube2l3 reduces pro-IL-1 β protein turnover.
7. In Figure S2A, MG132 and BAY 11-7082 treatments increased proIL-1 β abundance obviously as compared to DMSO or other treatment in iBMDMs, but in Figure 4A, the effects of these two inhibitors are not as pronounced as those of the former. Maybe the doxycycline-inducible pro-IL-1 β expression system is not necessary, and the LPS-induced endogenous pro-IL-1 β expression is also more realistic and easier.
8. In Figure 5A, silencing HERC3 seemingly had reduced the abundance of pro-IL-1 β rather than had no effect.
9. Many images of immunoprecipitation and immunoblot experiments are not marked accurately, this is difficult for readers to understand. Please check them carefully.
10. The coimmunoprecipitation of endogenous TRIP12 and AREL1 with UBE2L3 is necessary.
11. In the body of the article, there are many contents describing the experimental methods. These contents should be compressed and moved to figure legend, and more descriptions of the results should be added.
12. In Figure 6F, both pro-IL-1 β -4R and pro-IL-1 β -5R variants were remarkably more stable than wildtype pro-IL-1 β , but their half-lives are speculative rather than experimentally observed.
13. TRIP12 and AREL1 E3 ligases target pro-IL-1 β for ubiquitylation with K27-, K29- and K33- poly-ubiquitin chains, but the ubiquitin modification for proteasome degradation is K48- poly-ubiquitin chains rather than three of above, this is very strange. There should be other E3 ligases functioning on pro-IL-1 β to cause it degrade by proteasomes.

Minor points:

1. Flow cytometric analyses revealed fewer F4/80+ve macrophages staining positive for UBE2L3, and a reduction in the geometric mean fluorescence intensity for UBE2L3 with LPS or LPS and ATP, simultaneously, immunoblots showed lower UBE2L3 abundance in peritoneal macrophages from mice given LPS and ATP, what about the result for mice only given LPS?
2. In Figure 3E, why the abundance of pro-caspase-1 is higher in Ube2l3 Δ Mac iBMDMs than Ube2l3^{fx/fx}?
3. It is better to confirm the efficacy of silencing of Arel1, Trip12 and Herc3 by western blot rather than RT-qPCR.

Reviewer #3 (Remarks to the Author):

The reviewed manuscript by V. Mishra et al. reveals the mechanism of ubiquitination of IL1b, an important immune response cytokine, which involves ubiquitin conjugating enzyme UBE2L3 and two ubiquitin ligases TRIP12 and ARL1, that complete targeting of IL1b for proteasomal degradation. The work is an extension of their previous publication that identified UBE2L3 to be involved in IL1b ubiquitination. Given a central role of IL1b in host defense and in homeostasis, this is an important question, particularly given polymorphism in UBE2L3 linked to inflammatory conditions in humans. The presented data provide evidence of the physiological role of UBE2L3 in vivo and for the first time identify two Ub-ligases to be involved in regulation of the levels of IL1b. It is an impressive piece of work, which certainly advances the field, with my only concern related to the relatively technical nature of the discovery. Capitalizing on their data in making predictions to IL-1b biology would strengthen this undoubtedly important study.

- Line 18, "... caspase-1 inflammasomes" sounds vague.
- Line 20, "... actions in vivo and ubiquitin ligases..." sounds also disjointed.
- Line 85, how important is to know that UBE2L3 is regulated by caspase-1?
- Figure 1, panel A and further, the benefits of presenting dot-plot data combined in a shape similar to violin plot are not obvious. I would argue that it distracts rather than helps focusing on the data, but I will leave it to authors' discretion.
- The data in Figure 1 to some extent repeat what has been previously shown by the authors and therefore is perceived as relatively incremental.
- Figure 2 D-E, the data confirm that UBE2L3 doesn't target IL-6 rather than specifically target IL-1b. That is, it is fine to show IL-6 in comparison with IL-1b, but the interpretation should be different.
- Line 147, "... remarkably powerful role..." is vague.
- Line 155, once the gene-excision approach is introduced, mentioning it in the text could be omitted.
- Line 167, it is not clear why the immortalized macrophages are needed for the experiment.
- Line 179, the rationale for screen is relatively weak, and I wonder if it could be strengthened by investigating whether other processed by caspase-1 members of the IL-1b family cytokines could be also targets of the sought ligases/ UBE2L3? Again, the way it is presented in the paper, it looks more like a technical issue.
- Line 213, "family-wide" should be further specified.
- Line 213, why would the two ligases interact? Is this a scientifically valid question?
- In discussing the specific combination of E2 and E3 Ub-enzymes and in light of the comments above, is the established gene-combination confined to IL-1b only, or it could be extended to other IL-1b-family cytokines?
- What is the physiological relevance/importance of ubiquitination of the IL-1b precursor versus mature protein?
- In general, the Discussion should more open up the field rather than going over of how the data fit into what is known about IL-1b biology and processing.

We thank the reviewers for their comments. New text and figures are indicated in the point-to-point response below and in yellow in the main text.

Reviewer #1 (Remarks to the Author):

The manuscript by Mishra, Crespo-Puig et al. follows up on this group's published finding that the E2 ligase UbcH7 regulates pro-IL-1b stability. In the current work, the authors extend that Cell Reports paper to generate a Tamoxifen-inducible, myeloid-specific UbcH7 knockout mouse. They show that that mouse has increased IL-1b levels and release upon inflammasome activation with an increased half-life of pro-IL-1b. They further develop an siRNA screening assay to determine the E3 ubiquitin ligase(s) responsible for IL-1b ubiquitination. Using that system, they find that Are1 and Trip12 mediate IL-1b ubiquitination. The manuscript is logical, very well-written and really a joy to read. I think it will be of great interest to the innate immune and inflammasome communities. There are some points and experimentation that need clarification and/or extension, however. My specific comments are as follow:

1. In Figure 1G, it isn't clear why UbcH7 mRNA levels fall in response to LPS + ATP. This could simply be an artifact of the cell undergoing pyroptosis. This needs to be examined further, perhaps with a GasderminD inhibitor, LPS and ATP alone and/or Gasdermin D knockout cells.

Fig 1G did not show RT-PCR data; it showed UBE2L3 quantification from western blots in the old Fig 1F. To make this clearer, and in response to reviewer #2 (who requested data from LPS-treated mice alongside other treatments), we have replaced both panels with new data from mice treated with PBS, LPS and LPS+ATP. Quantification is now indicated below the lanes in Fig 1F.

2. It isn't clear from the manuscript if the UbcH7-knockout myeloid mice have an inflammatory phenotype or inflammatory pathology. What does a necropsy show? Along these same lines, the authors are injecting LPS into these mice. Did the authors do an LPS-sepsis study? Presumably the UbcH7 mice would show lethality at a lower dose of LPS?

Inflammatory phenotype vs pathology is an interesting question as excessive inflammation over long periods of time leads to pathology. We also wish to point out that 30mg/kg is a lower dose than 50-60 mg/kg that is often used in sepsis models, and that at this dose UBE2L3-KO mice show much higher disease scores and elevated IL-1 β (Fig 2H). In the LPS-induced sepsis-like model, high IL-1 β induces neutrophilic influx and pathology in several tissues, including the lung and liver at late time points (> 24 h). However, we do

not have permission to perform a sepsis study (i.e., time-to-death) under our PPL because of the high severity and ethical concerns of excessive pain and discomfort to a sentient species such as mice. Therefore, late time points are not possible. Nonetheless, as this reviewer suggested, we performed H&E staining of lung sections from LPS treated WT & UBE2L3-KO mice at 3 h. As we expected at this early time point, blinded scoring by an expert pathologist revealed no differences in the pathology between two genotypes even though, 'qualitatively' we did observe higher neutrophilic influx in UBE2L3-KO lungs. Images below are one randomly selected field of view for WT and KO mice (n = 3 each), and blinded scores of the H&E staining.

3. Figure 3E would be more convincing with a time course of IL-1 β release and cleavage as well as a time course of Gasdermin D cleavage. It isn't clear until Figure 5I that pyroptosis (and presumably Gasdermin D cleavage) is unaffected in the UbcH7-null mice.

We agree that we should have emphasised the lack of an impact on pyroptosis with loss-of-function of UBE2L3, TRIP12 and ARL1 better. We have now included a time course of IL-1 β showing higher release by Ube2l3-KO primary BMDMs and similar pyroptosis and IL-6 over time. We also show western blotting from primary BMDMs (as requested by reviewer #3) showing similar gasdermin-D cleavage and caspase-1 activation in WT and Ube2l3-KO cells. Together, this reaffirms unaffected pyroptosis in UbcH7-null mice. These data are in new Fig 3E-H as shown below:

Fig 3
These results are described in lines 164-176 in the text:

"In line with high pro-IL-1 β levels, NLRP3 inflammasome activation with LPS and nigericin resulted in prominently higher mature IL-1 β as measured by immunoblots and ELISA in Ube2l3 Δ Mac primary BMDMs (Figure 3E-F). Importantly, caspase-1 activation into its p20 form and gasdermin-D cleavage were similar in both genotypes (Figure 3E), ruling out impact on inflammasome activation or proteolysis of another key substrate of caspase-1. In agreement with this, IL-6 release and pyroptotic cell death as measured by propidium iodide dye uptake were also indistinguishable between the two genotypes (Figure 3G-H). These results are indicative of normal NF- κ B signalling, inflammasome priming and activation and pyroptotic cell death in Ube2l3 Δ Mac cells. We therefore conclude that deletion of Ube2l3 reduces pro-IL-1 β protein turnover but, it does not affect inflammasome activation or pyroptosis. Taken together, these results from knockout mice unequivocally establish a crucial role for UBE2L3 in acting independently of inflammasome activation in reducing pro-IL-1 β turnover."

4. It should be at least mentioned in Figure 4A that BAY11-7082 is an IKK inhibitor. It won't matter here as they aren't looking at endogenous pro-IL-1 β , but NF- κ B inhibition could be affecting their system and it should at least be mentioned in the text.

The reviewer is correct, we have added this to line 184 as follows:

"Firstly, BAY-11-7082, which can inhibit UBE2L3 and RBR E3 ligases within the linear ubiquitin assembly complex (LUBAC) 13, among other targets including IKK, increased pro-IL-1 β abundance as compared to vehicle (DMSO)-treated cells (Figure S2A)."

5. Not a critique, but I find the siRNA screen to be elegant and well-thought out with very good validation. It wasn't clear to me as a reader, however, how the authors settled on HECT/RBR E3 ligases. I'm assuming that these are the ones known to use Ubch7, but a sentence or two in the manuscript formally explaining this would be useful.

We thank the reviewer for their positive comments. We apologise that we did not make the rationale sufficiently clear. Lines 209-216 state this in the following way:

“UBE2L3 is a unique E2 enzyme that can only transfer the ubiquitin from its active-site to cysteine residues in the active sites of E3 ligases, and unlike other E2 enzymes it cannot discharge ubiquitin from its active site directly to lysine residues on target proteins ⁹. Therefore, it only cooperates with HECT and RBRs E3 ligases which ubiquitylate proteins by first transferring ubiquitin to their own active site cysteine residue; in contrast, RING ligases directly transfer ubiquitin from their cognate E2 to lysine on substrates. Therefore, family-wide siRNA screening was performed against 43 E3-ligases of the HECT and RBR families which to identify E3 ligases involved in pro-IL-1 β ^{strep} turnover...”

6. In Figure 5 and 6, it isn't clear the relative contributions of TRIP12 and Arel1 to the total phenotype of the Ubch7-null cells. That is, what happens when you compare levels of IL-1b in when you use siRNA to knockdown Ubch7, Trip12 and Arel1 in the same system? Do these two E3 ligases mediate all of the Ubch7 effect? Are they synergistic or parallel?

Thank you for this insightful question – we have performed new experiments to address this. Firstly, the combined silencing Trip12 and Arel1 led to higher pro-IL-1 β abundance than their silencing each of them alone (new Fig S2H; lines 228-231). This indicates that both E3 ligases have a part to play in parallel for complete pro-IL-1 β turnover working with UBE2L3 as the E2 partner. Secondly, we have addressed this by silencing Trip12 or Arel1 in Ube2l3-KO macrophages, which showed that, unlike in WT BMDMs, pro-IL-1 β abundance does not further increase upon silencing of these E3 ligases in UBE2L3-KO cells (new Fig 5G). This indicates that UBE2L3 is responsible for their combined actions and other E2 cannot be used in this process. Taken together with our finding that TRIP12 and AREL1 form a complex, our findings indicate that they partner with UBE2L3 for modifying pro-IL-1 β with unusual ubiquitin chains to destabilise pro-IL-1 β (lines 241-250).

7. The authors should be commended for using denaturing conditions in their ubiquitination experiments in Figure 6. Not doing this is a common cause of ubiquitination artifacts and this attention to detail is quite nice to see.

We thank the reviewer for their praise and agree that this is less common than it should be.

In summary, I find the work as a whole to be well done and extremely interesting. I think attention to the concerns raised above would improve the manuscript, but overall, I quite enjoyed reading it.

Reviewer #2 (Remarks to the Author):

In this study, it is reported that deletion of Ube2l3 in mice markedly reduces pro-IL-1 β turnover in macrophages, leading to excessive mature IL-1 β production, neutrophilic inflammation and disease symptoms following inflammasome activation. The authors use a family-wide siRNA screen identified two ubiquitin ligases, TRIP12 and AREL1, which add K27-, K29- and K33- poly-ubiquitin chains on lysine residues in the 'pro' domain and destabilise pro-IL-1 β . Mutation of ubiquitylation sites increased pro-IL-1 β stability, but did not affect proteolysis by caspase-1. The extent of mature IL-1 β production is determined by precursor abundance, and UBE2L3, TRIP12 and AREL1 limit inflammation by shrinking the cellular pool of pro-IL-1 β . This study uncovered fundamental processes governing IL-1 β homeostasis and provided molecular insights that could be exploited to mitigate its adverse actions in disease. However, this article still has many shortcomings to be improved.

Major points:

1. The article called "The Atypical Ubiquitin E2 Conjugase UBE2L3 Is an Indirect Caspase-1 Target and Controls IL-1 β Secretion by Inflammasomes" (DOI: 10.1016/j.celrep.2017.01.015) has already revealed that UBE2L3 acts post-translationally to promote K48-ubiquitylation and turnover of pro-IL-1 β and dampen mature-IL-1 β production and UBE2L3 depletion increases pro-IL-1 β levels and mature-IL-1 β secretion by inflammasomes, so much of the work in this article seems to be just repetition and further corroborate this conclusion.

We respectfully disagree that any of the work shown here is a repetition. All experiments and reagents described here are new: 1. new UBE2L3 knockout mice, 2. In vivo experiments showing UBE2L3 depletion (previous work was entirely *in vitro*), 3. new unbiased RNAi screen that identified two E3 ligases that turnover pro-IL-1 β , 4. Characterization of ubiquitin chains and residues in IL-1 β that are modified, 5. the impact of ubiquitylation/lack thereof on IL-1 β maturation. This builds substantially on our previous work in which we mainly focused on the discovery of UBE2L3 as an indirect target of inflammasomes in mouse and human macrophages. Although we also identified UBE2L3 promotes IL-1 β turnover, the precise mechanisms, E3 ligases involved and the role of UBE2L3 in vivo were not investigated. More importantly, the new knockout and in vivo experiments agree with previous finding and strengthen this manuscript and robustness of our findings.

2. Mice were given PBS, LPS (5 μ g, 3 h) or LPS+ATP (5 μ g, 3 h + 50 μ mol, 10 min) intraperitoneally, it seems that the dose should be based on the weight of the mouse, rather than simply how much reagent to give a mouse.

We apologise for not clarifying this – this is a previously established model ("Nizami, Sohaib, et al. "Inhibition of the NLRP3 inflammasome by HSP90 inhibitors." *Immunology* 162.1 (2021): 84-91"). We agree with the reviewer that the dose should be normalised to weight of mice. Therefore, we used mice of 21-22 g weight so the doses of LPS and ATP are comparable. This is clarified in the Methods in Lines 806-7.

“Age and sex matched mice weighing 20-21 g were given LPS (5 µg in sterile PBS) intraperitoneally for 3 h followed by an intraperitoneal injection of 50 µmol of ATP (in sterile PBS).”

3. On line 110, the paper says that UBE2L3 could act as a negative regulator of IL-1β production in vivo, but the above data alone does not seem to lead to such a conclusion.

We do not claim that this data alone led us to this conclusion – we have said “Taken together... UBE2L3 *could* act as a negative regulator”, which provides rationale for further experiments in the paper. We have reworded this sentence as follows (lines 110-11):

“These results from experiments in vivo and previous findings in vitro ⁶ are consistent with UBE2L3 acting as a negative regulator of IL-1β production.”

4. In Figure S1, the result of PCR-based screening for genotyping WT (*Ube2l3^{fx/fx}*) and KO (*Ube2l3^{ΔMac}*) primary BMDMs given 4-hydroxytamoxifen (2 µM) for 48 h is inconsistent with Oligonucleotide primer binding sites indicated.

We apologise for this and have labelled all oligos and provided a clearer legend to fig S1A. The crux of the screening strategy (developed and provided by Taconic) is this: oligos 1-4 PCR only works when the long intervening genomic DNA has been ‘floxed’ out; oligo 1-2 are closer and work in WT and Tam-treated KO showing residual WT allele. This is explained in the legend to Fig S1A.

“(A) Schematic depiction (left) and PCR-based screening (right) for genotyping of primary BMDMs from WT (*Ube2l3^{fx/fx}*) and KO (*Ube2l3^{fx/fx}/cre/Esr1* mice labelled *Ube2l3^{ΔMac}* for convenience throughout) mice. Oligonucleotide (oligo) primer binding sites are indicated. Oligo 2 binding site is lost after Cre-mediated deletion of exon 1. Cells were treated with 4-hydroxytamoxifen (hTam, 2 µM) for 48 h before genomic DNA was prepared for PCR. Oligos 1 and 4 only generate a product if homologous recombination has occurred in cells expressing Cre (i.e., cells from KO mice), and no amplification is observed without Cre recombinase (i.e., WT cells). Oligos 1 and 2 generate a product at the “WT” locus in both genotypes, which indicates remnant WT alleles after hTam treatment in cells from KO mice.”

5. In Figure 2, the data of three established models of inflammasome—dependent inflammation and quantified IL-1β production and inflammatory parameters has a surprisingly large margin of error in each experiment group, what do you think caused this and how to improve this?

Thank you for this question. Firstly, when plotted ‘as commonly done’ as bar±SEM, the differences may perhaps be more apparent as these kinds of plots are more common in literature. The panels below are the same data shown in Fig 2D-H.

However, in the interest of transparency and accuracy we have showed all data from all experiments in box/whiskers and violins in the manuscript. The underlying variability could be due to multiple reasons. We had to perform experiments over a period of >2y as we had to nearly close the mouse colony during the pandemic and build it up again afterwards. Secondly, we have shown data from all experiments (4 different occasions with ~4 mice per group, with different batches of LPS and ATP). Avoiding this, where possible, could reduce variance. We also designed experiments as randomised blocks and took into account the inter-experiment variability by analysing data using mixed effects ANOVAs (mixed effects linear models). This design increases the power of detecting the effect of “Genotype” while reducing the number of animals needed and is consistent with principles of 3R.

6. On line 175-176, the author said that these results from knockout mice unequivocally establish a crucial role for UBE2L3 in acting ‘upstream’ of inflammasomes by reducing pro-IL-1β abundance, this is questionable and the results only indicate that deletion of Ube2l3 reduces pro-IL-1β protein turnover.

Our experiments taken together indicate that the reduced turnover increases pro-IL-1β abundance. We have reworded this particular sentence (lines 182-184) as follows: “Importantly, these results from knockout mice unequivocally establish a crucial role for UBE2L3 in acting independently of inflammasome activation in reducing pro-IL-1β turnover.”

7. In Figure S2A, MG132 and BAY 11-7082 treatments increased proIL-1β abundance obviously as compared to DMSO or other treatment in iBMDMs, but in Figure 4A, the effects of these two inhibitors are not as pronounced as those of the former. Maybe the doxycycline-inducible pro-IL-1β expression system is not necessary, and the LPS-induced endogenous pro-IL-1β expression is also more realistic and easier.

This is an astute observation. We were aware of the risks of a poor assay leading to hits that need secondary screens to weed out uninteresting/irrelevant hits based on our

previous experience with RNAi screens (PMCID: PMC5493756). We therefore tested different approaches (not shown) before settling on the Dox-inducible system which successfully led to the discovery of E3 ligases of interest. Alternatives that we considered were CMV or other promoter-driven constitutive expression of pro-IL-1 β ; however, the screen would then require a 'cycloheximide chase', which was technically more challenging. Even though the dynamic range with the Dox setup is lower than that of endogenous pro-IL1 β induced by LPS, there are significant advantages of this approach. The main one is that this system allowed us to follow protein abundance without the confounding effects of IL-1 β transcription that is by LPS via NF- κ B in macrophages. Second, we avoided having to perform a secondary siRNA screen to weed-out hits that affect NF- κ B—driven transcription rather than protein turnover.

8. In Figure 5A, silencing HERC3 seemingly had reduced the abundance of pro-IL-1 β rather than had no effect.

This was not consistent across independent experiments (e.g., see quantification across experiments in Fig 5B), even though it appears to be slightly reduced in this experiment. We therefore did not follow up HERC3.

9. Many images of immunoprecipitation and immunoblot experiments are not marked accurately, this is difficult for readers to understand. Please check them carefully.

We apologise sincerely for this oversight. We have relabelled all blots which should now be free of errors. Particularly, we clarify in figure legends that the anti-FP (fluorescent protein) antibody reacts with GFP and Venus tagged proteins.

10. The coimmunoprecipitation of endogenous TRIP12 and AREL1 with UBE2L3 is necessary.

We apologise that we have been unable to do this despite trying 3 different antibodies each against human TRIP12 and AREL1. No reliable commercial antibodies are available against murine proteins.

However, we had shown that transfected GFP-TRIP12 and Venus-AREL1 co-immunoprecipitate endogenous UBE2L3 (Fig 5D).

Moreover, the interaction between these pairs of E2/E3 has been established previously: TRIP12 & UBE2L3 (Gatti, 2019, Cell Rep, PMID 32755579; Kaiho-Soma, Mol Cell, PMID 33567268; Pao, 2017, Nature, PMID 29643511) and AREL1 & UBE2L3 (Schwarz, 1998, J Biol Chem, PMID 9575161; Kristariyanto, 2015, Biochem J, PMID 25723849; Lear, 2016, J Exp Med, PMID 27162139; Michel, 2015, Cell, PMID 25752577). Here we have established their function in IL-1 β turnover.

11. In the body of the article, there are many contents describing the experimental methods. These contents should be compressed and moved to figure legend, and more descriptions of the results should be added.

Thank you for this suggestion, we have changed text to simplify it – especially in sections describing the generation of KO mice and aspects related to the siRNA screen. We had included this to make it easier without having to refer to Methods repeatedly.

12. In Figure 6F, both pro-IL-1 β -4R and pro-IL-1 β -5R variants were remarkably more stable than wildtype pro-IL-1 β , but their half-lives are speculative rather than experimentally observed.

We are sorry we did not clarify that these half-lives were experimentally derived similarly to that of endogenous pro-IL-1 β described in Fig 3A-B. The graphs showing this is now shown in Fig S3E.

13. TRIP12 and AREL1 E3 ligases target pro-IL-1 β for ubiquitylation with K27-, K29- and K33- poly-ubiquitin chains, but the ubiquitin modification for proteasome degradation is K48- poly-ubiquitin chains rather than three of above, this is very strange. There should be other E3 ligases functioning on pro-IL-1 β to cause it degrade by proteasomes.

This is not strange given recent work that has showed K11, K27 and K29 can all target proteins for degradation, and that additional K48 chains can be added on these ‘base’ chains. This is known for TRIP12 (Kaiho-Soma et al, Mol Cell, PMID: 33567268; Khan OM et al, Nat Commun, PMID: 33824312) and other substrates (Tracz & Bialek, Cell Mol Biol Lett, PMID: 33402098). The absence of Ube2l3/Trip12/Arel1, which add these unusual chain types, markedly slows pro-IL-1 β turnover indicating these chains are crucial. We had already mentioned in the Discussion that future work should identify the E3 ligases that may add K48 chains (lines 372-80):

“As TRIP12 and AREL1 did not modify pro-IL-1 β with K48- or K11- ubiquitin chains, is plausible that additional E3 ligases participate in pro-IL-1 β ubiquitylation with other chain types following the initial ubiquitin chains added by TRIP12-AREL1-UBE2L3. Interestingly, silencing TRIP12 or AREL1 in *Ube2l3* ^{Δ Mac} cells did not further impact pro-IL-1 β abundance, which indicates that other E2 enzymes cannot compensate for loss of UBE2L3. Future work should clarify whether UBE2L3 can promote pro-IL-1 β ubiquitylation via E3 ligases that deploy unusual ubiquitylation mechanisms, such as the RING-cysteine-relay, RING-Zn finger mechanisms or a ‘super assembly’ of different E3 ligases ⁵¹.”

Minor points:

1. Flow cytometric analyses revealed fewer F4/80+ve macrophages staining positive for UBE2L3,

and a reduction in the geometric mean fluorescence intensity for UBE2L3 with LPS or LPS and ATP, simultaneously, immunoblots showed lower UBE2L3 abundance in peritoneal macrophages from mice given LPS and ATP, what about the result for mice only given LPS?

New data with all three treatments are now shown in Fig 1F, which are consistent with the flow cytometry analyses.

Fig 1F

2. In Figure 3E, why the abundance of pro-caspase-1 is higher in Ube2l3 Δ Mac iBMDMs than Ube2l3^{fx/fx}?

We did not observe markedly higher pro-casp1 across experiments and this is not Ube2l3-deletion dependent. This data has been replaced with new data from primary BMDMs (as asked by reviewer #3), which show similar levels of active caspase-1 and cleaved GSDMD in WT and KO cells.

3. It is better to confirm the efficacy of silencing of Arel1, Trip12 and Herc3 by western blot rather than RT-qPCR.

We agree, however, we are limited by reagents. However, we show siRNA efficacy indirectly using tagged proteins (GFP-TRIP12 and Venus-AREL1) whose abundance reduced markedly following siRNA treatment, confirming that the siRNA are effective in reducing protein abundance (even when overexpressed through the strong CMV-promoter plasmids). These new data are in Fig S2F-G

Fig S2

Reviewer #3 (Remarks to the Author):

The reviewed manuscript by V. Mishra et al. reveals the mechanism of ubiquitination of IL1b, an important immune response cytokine, which involves ubiquitin conjugating enzyme UBE2L3 and two ubiquitin ligases TRIP12 and ARL1, that complete targeting of IL1b for proteasomal degradation. The work is an extension of their previous publication that identified UBE2L3 to be involved in IL1b ubiquitination. Given a central role of IL1b in host defense and in homeostasis, this is an important question, particularly given polymorphism in UBE2L3 linked to inflammatory conditions in humans. The presented data provide evidence of the physiological role of UBE2L3 *in vivo* and for the first time identify two Ub-ligases to be involved in regulation of the levels of IL1b. It is an impressive piece of work, which certainly advances the field, with my only concern related to the relatively technical nature of the discovery. Capitalizing on their data in making predictions to IL-1 β biology would strengthen this undoubtedly important study.

We thank the reviewer for these positive comments, and agree that these findings have important implications on IL-1 β biology. However, we disagree that our findings are 'merely' technical as the identification of E3 ligases in the process and the generation and studies in new KO mice substantially add to our knowledge in this area.

- Line 18, "... caspase-1 inflammasomes" sounds vague.

We have changed this to "maturation of the pro-IL-1 β precursor by caspase-1". We have also rewritten the abstract to be under 150 words as per journal instructions.

- Line 20, "... actions *in vivo* and ubiquitin ligases..." sounds also disjointed.

We have reworded this to: "However, actions of UBE2L3 *in vivo* and its ubiquitin ligase partners in this process are unknown." We have also rewritten the abstract to be under 150 words as per journal instructions.

- Line 85, how important is to know that UBE2L3 is regulated by caspase-1?

This is an interesting question. Although strictly speaking UBE2L3 activity may be regulated (reduced) by different processes (e.g., caspase-1, transcription), the net effect of reduction in UBE2L3 would still be higher pro-IL-1 β abundance. What is interesting, however, is that one target of caspase-1 (i.e., UBE2L3) regulates another important target (i.e., pro-IL-1 β). Further studies will be needed to identify the collective actions of caspase-targets.

- Figure 1, panel A and further, the benefits of presenting dot-plot data combined in a shape similar to violin plot are not obvious. I would argue that it distracts rather than helps focusing on the data, but I will leave it to authors' discretion.

We thank the reviewer for letting us decide. Our overall goal when developing and using these types of graphs was to show all data points and the inherent biological variability in

a transparent manner. However, we have taken this reviewer's views onboard. We have increased the visibility of the boxplots relative to that of the violins and scattered data symbols so that differences between groups are more easily discernible. We think they may take more getting used to as they are uncommon in our fields.

- The data in Figure 1 to some extent repeat what has been previously shown by the authors and therefore is perceived as relatively incremental.

We respectfully disagree this is a repeat or incremental – we (or others) have not previously shown UBE2L3 depletion in vivo. We therefore feel it adds to the findings on UBE2L3 as a key E2 conjugating enzyme and biologically supports the idea of it acting as a negative regulator of IL-1 β production in vivo. Given the association of disease-linked polymorphisms in UBE2L3 in humans (see review by Alpi et al, cited, PMID: 27729585), we believe these in vivo data are important for the field of ubiquitylation in general – not just inflammasomes/caspase-1 and IL-1.

- Figure 2 D-E, the data confirm that UBE2L3 doesn't target IL-6 rather than specifically target IL-1b. That is, it is fine to show IL-6 in comparison with IL-1b, but the interpretation should be different.

We agree, and have reworded this to “indicating a lack of effect on an inflammasome-independent cytokine upon deletion of *Ube2l3*” (lines 129-30).

- Line 147, “... remarkably powerful role...” is vague.

This wording is based on the effect of UBE2L3 deletion in one compartment leading to such effects, showing its impact. We have reworded this to “revealing its **important** role in limiting IL-1 β —driven inflammation.” (line 147).

- Line 155, once the gene-excision approach is introduced, mentioning it in the text could be omitted.

We agree, and have changed this throughout – including the generation of KO and aspects related to the siRNA screen.

- Line 167, it is not clear why the immortalized macrophages are needed for the experiment.

This was done to reduce the number of mice needed and we had to reduce our colony through the pandemic. These data have been replaced with results from primary BMDMs, which are consistent with those from iBMDMs (new Fig 3E-H). These results are described in lines 164-176.

Fig 3

• Line 179, the rationale for screen is relatively weak, and I wonder if it could be strengthened by investigating whether other processed by caspase-1 members of the IL-1 β family cytokines could be also targets of the sought ligases/ UBE2L3? Again, the way it is presented in the paper, it looks more like a technical issue.

We respectfully disagree that the screen has a weak rationale – reviewer 1 said “...I find the siRNA screen to be elegant and well-thought out with very good validation”. E3 ligases determine protein turnover, and their identification can open ways to target them therapeutically in the future. This is already being attempted in oncology. With > 700 E3 ligases in the genome, the screen had to be designed carefully. The screen was unbiased and therefore a powerful way to find new proteins involved in the process. Our strategy, which did take time to develop, avoids the need for secondary screens (e.g., hits to irrelevant processes governing IL-1 β abundance). Moreover, we are now able to use the same strategy to identify deubiquitylases, small molecule regulators of the process. We apologise our writing made it sound ‘technical’, and we have changed this in the revision. A deeper analyses of other IL-1 family members (there are 8 and not all are caspase-1 substrates) is beyond the scope of this revision, but it is a topic of interest.

• Line 213, “family-wide” should be further specified.

We apologise this was not clarified – we have mentioned this on lines 209-216

“UBE2L3 is a unique E2 enzyme that can only transfer the ubiquitin linked to its own active-site to cysteine residues in the active sites of E3 ligases, and unlike other E2 enzymes it cannot discharge ubiquitin from its active site directly to lysine residues on target proteins⁹. Only HECT and RBRs E3 ligases catalyse target ubiquitylation via such a reaction intermediate, whereas RING ligases directly transfer ubiquitin from their cognate E2 to lysine on substrates. Therefore, family-wide siRNA screening was performed against 43 E3-ligases of the HECT and RBR family which are known to cooperate with UBE2L3 to determine their role in the turnover of pro-IL-1 β ^{strep}.”

- Line 213, why would the two ligases interact? Is this a scientifically valid question?

It is an important question as E3 ligases often work along with other E3 ligases in complexes (e.g., the best studied are BRCA1—BARD1 and MDM2—MDMX heterodimeric E3 ligases in cancer). While these are RING ligases, recent structural and biochemical studies have revealed a “super assembly” of the RBR E3 ligase ARIH1 and UBE2L3, along with a Cullin-RING ligase in cooperatively ubiquitylating substrates (Horn-Ghetko et al, Nature, PMID: 33536622). Similar interaction interfaces are present in the subunits of LUBAC (i.e., HOIP (catalytic RBR component) and HOIL1 (structurally related but not catalytically active) Balaji et al, F1000Res, PMID: 32076548). Ours is the first report of the large E3 ligases TRIP12 and AREL1 collaborating in this way, and we are confident will be of much interest in the field of ubiquitylation in general.

- In discussing the specific combination of E2 and E3 Ub-enzymes and in light of the comments above, is the established gene-combination confined to IL-1b only, or it could be extended to other IL-1b-family cytokines?

We are currently investigating the ubiquitylation of the entire IL1 family of human and mouse proteins, but this is beyond the scope of this study. We wish to point out that the long pro-domain is only present in IL-1 α and IL-1 β and the K133 residue in the mature region is only conserved in IL-1 β , IL-37 and IL-38. Therefore, related yet distinct principles apply to other family members. We have mentioned this in the Discussion: lines 397-401

“The N-terminal ‘pro’ domain is unique to pro-IL-1 α and pro-IL-1 β , and K133 is conserved only some of the IL-1 family members (i.e., IL1RA, IL-37 and IL-38). An interesting question to address in the future is the advantage of ubiquitylation in the pro-domain versus the mature domain, and the contribution of TRIP12, AREL1 and UBE2L3 in regulating other IL-1 family members.”

- What is the physiological relevance/importance of ubiquitination of the IL-1b precursor versus mature protein?

This is an interesting question. Ubiquitylation of the mature domain will require that it be removed prior to secretion, whereas if in the pro-domain it will be removed by cleavage by caspase-1 to generate the mature, bio-active form. Given that this region is unique to some IL-1 family members, we presume it has additional roles that remain to be identified. Here we have attributed a previously unappreciated role for the pro-domain in controlling protein stability.

- In general, the Discussion should more open up the field rather than going over of how the data fit into what is known about IL-1b biology and processing.

Thank you for this suggestion, we have done this. The current Discussion did cover topics such as the disease-relevance of UBE2L3, its pairing with bacterial pathogens, that unusual ubiquitin chains can target several other proteins for disposal, and PROTACs are an

interesting avenue to explore. We limited further speculation based on previous experience with some reviewers wanting us to avoid this. Further, we feel this balance of topics is better suited to the general readership of this journal.

We have added the following on E3 ligase complexes and the combined actions of TRIP12/AREL1/UBE2L3 (lines 372-380):

“As TRIP12 and AREL1 did not modify pro-IL-1 β with K48- or K11- ubiquitin chains, is plausible that additional E3 ligases participate in pro-IL-1 β ubiquitylation with other chain types following the initial ubiquitin chains added by TRIP12-AREL1-UBE2L3. Interestingly, silencing TRIP12 or AREL1 in *Ube2l3* Δ^{Mac} cells did not further impact pro-IL-1 β abundance, which indicates that other E2 enzymes cannot compensate for loss of UBE2L3. Future work should clarify whether UBE2L3 can promote pro-IL-1 β ubiquitylation via E3 ligases that deploy unusual ubiquitylation mechanisms, such as the RING-cysteine-relay, RING-Zn finger mechanisms or a ‘super assembly’ of different E3 ligases 51. ”

We have added the following on IL-1 family (lines 397-401):

“The N-terminal ‘pro’ domain is unique to pro-IL-1 α and pro-IL-1 β , and K133 is conserved only some of the IL-1 family members (i.e., IL1RA, IL-37 and IL-38). An interesting question to address in the future is the advantage of ubiquitylation in the pro-domain versus the mature domain, and the contribution of TRIP12, AREL1 and UBE2L3 in regulating other IL-1 family members.”

REVIEWERS' COMMENTS

Reviewer #1 (Remarks to the Author):

The authors have satisfactorily satisfied my critiques.

Reviewer #2 (Remarks to the Author):

I have no more concerns.

Reviewer #3 (Remarks to the Author):

I have read the revised manuscript to find it significantly improved. I have no further comments

Response to reviewers' comments.

Reviewer #1 (Remarks to the Author):

The authors have satisfactorily satisfied my critiques.

Reviewer #2 (Remarks to the Author):

I have no more concerns.

Reviewer #3 (Remarks to the Author):

I have read the revised manuscript to find it significantly improved. I have no further comments

We thank the reviewers for their comments which helped improve the manuscript.